# Class Probability Matching with Calibrated Networks for Label Shift Adaptation

**Hongwei Wen, Annika Betken & Hanyuan Hang**
Faculty of Electrical Engineering, Mathematics and Computer Science
University of Twente
Enschede, The Netherlands
`{h.wen,a.betken,h.hang}@utwente.nl`

## Abstract

We consider the domain adaptation problem in the context of label shift, where the label distributions between source and target domain differ, but the conditional distributions of features given the label are the same. To solve the label shift adaptation problem, we develop a novel matching framework named *class probability matching* (*CPM*). It is inspired by a new understanding of the source domain's class probability, as well as a specific relationship between class probability ratios and feature probability ratios between the source and target domains. CPM is able to maintain the same theoretical guarantees as the existing feature probability matching framework, while significantly improving the computational efficiency due to directly matching the probabilities of the label variable. Within the CPM framework, we propose an algorithm named *class probability matching with calibrated networks* (*CPMCN*) for target domain classification. From the theoretical perspective, we establish a generalization bound of the CPMCN method in order to explain the benefits of introducing calibrated networks. From the experimental perspective, real data comparisons show that CPMCN outperforms existing matching-based and EM-based algorithms.

## 1 Introduction

In classification problems, domain adaptation refers to the ability of an algorithm trained on a source domain with distribution $P$ to successfully fulfill its task in a different, but related, target domain with distribution $Q$. As one of the most commonly used domain shift assumptions, label shift (Saerens et al., 2002; Storkey, 2009) assumes that the class probability varies, i.e. $p(y) \neq q(y)$, while the class-conditional probability remains unchanged, i.e. $p(x|y) = q(x|y)$. Domain adaptation under the label shift assumption has been utilized in a wide range of real-world applications, including disease diagnoses (Zhang et al., 2013; Tasche, 2017), visual recognition (Martinez et al., 2017; Yang et al., 2018) and relation extraction (Hong et al., 2022).

One paradigm to solve the label shift adaptation problem are matching methods. An example is Zhang et al. (2013). This article extends the kernel mean matching method from Gretton et al. (2009) to label shift settings for estimating the class probability $q(y)$, assigns weights for samples according to the class probability ratio $q(y)/p(y)$, and retrains a weighted classifier in the source domain that can be used in the target domain. More recently, network-based methods (Guo et al., 2020) achieves high performance via training generative adversarial networks (GANs). These methods are based on feature probability matching on $X$. They come with strong theoretical guarantees for the solution, but suffer from huge computational costs. To reduce the computational complexity, researchers have sought to match the probabilities of the prediction $\widehat{Y}$ instead of matching the probabilities of the feature $X$. To this end, Lipton et al. (2018); Azizzadenesheli et al. (2019) train a classifier $\widehat{h}$ on the source domain to obtain predicted labels $\widehat{y} = \widehat{h}(x)$ in the target domain, and then match the estimated target class probability $q(\widehat{y})$. Recently, motivated by semiparametric models, Tian et al. (2023) match the moment of a variable related to the prediction $\widehat{y}$. However, the cost of these methods is that the variable $\widehat{y}$, on which the probability matching is conducted, suffers from prediction errors of the fitted classifier $\widehat{h}$.

Another paradigm to address the label shift adaptation problem are EM-based algorithms. For instance, Chan & Ng (2005) propose an EM-based algorithm inspired by Saerens et al. (2002) to estimate $q(y)$. A corresponding classifier then selects the class label with the maximal estimated probability $q(y|x)$. Furthermore, Storkey (2009) assumes a prior over $p(y)$ and estimates $q(y)$ using Bayes' Rule, while Gretton et al. (2012) combine anti-causal learning with the label shift assumption to compute $q(y)$. However, for most of these methods, the use of intermediate variables like $p(x|y)$ or $p(x)$, which themselves are difficult to estimate, hinders the estimation performance of the predictor $q(y|x)$. Their further development has been limited until Alexandari et al. (2020), which combines the calibrated neural networks (Platt, 1999; Guo et al., 2017) and EM methods to estimate $q(y)$ and $q(y|x)$. As modern neural networks are no longer well-calibrated, calibration techniques are introduced to enable estimated conditional probabilities to accurately reflect the true likelihood of correctness (Guo et al., 2017). Experimentally, Alexandari et al. (2020) demonstrate that without calibration, EM-based algorithms perform worse than matching methods, while after calibration, EM-based algorithms can outperform matching methods, validating the effectiveness of calibrated networks in label shift adaptation problems.

In this paper, we propose a novel matching framework named *class probability matching*, which is based on matching the distributions of the label $Y$. Specifically, we provide new theoretical understanding on the source domain's class probability $p(y)$, as well as a specific relation between the class probability ratio $q(y)/p(y)$ and the feature probability ratio $p(x)/q(x)$ under the label shift assumption. This results in a new matching framework of two probabilities on the label variable $Y$ to get the class probability ratio. Accordingly, we propose a novel algorithm named *class probability matching with calibrated networks* (*CPMCN*) for label shift adaptation. Our approach consists of estimating the class probability ratio with the use of calibrated neural networks and obtaining the classifier for the target domain. We provide rigorous theoretical guarantees for the solution of our novel class matching framework on $Y$, as well as the generalization bound of our proposed CPMCN to demonstrate the benefit of introducing a calibrated network. The proposed algorithm is not only computationally more efficient than the matching methods on the feature vector $X$, but also avoids introducing an additional prediction error in contrast to matching methods predicting $\widehat{Y}$. The contributions of this paper are summarized as follows.

*(i)* To solve the label shift adaptation problem, we develop a novel matching framework named class probability matching that directly matches on the probabilities of label $Y$. Based on this framework we propose a new algorithm called CPMCN for label shift adaptation, which applies the calibrated neural network. CPMCN has low computational complexity and high theoretical guarantees.

*(ii)* Theoretically, we provide rigorous theoretical guarantees for our proposed matching framework. Moreover, we establish a generalization bound for the CPMCN algorithm, which illustrates the benefit of incorporating a calibrated network in the algorithm.

*(iii)* Experimentally, we validate that CPMCN outperforms existing matching methods and EM-based methods, in class probability ratio estimation and target domain classification.

## 2 PRELIMINARIES

### 2.1 DOMAIN ADAPTATION

In domain adaptation for classification, the input space $\mathcal{X} \subset \mathbb{R}^d$ and output space $\mathcal{Y} = [K] := \{1, 2, \ldots, K\}$ of both, source and target domain, are the same, but the distributions of source and target data, i.e. the corresponding probability distribution $P$ and $Q$ on $\mathcal{X} \times \mathcal{Y}$, differ. In domain adaptation, we observe labeled source domain data $D_p := \{(X_1, Y_1), \ldots, (X_{n_p}, Y_{n_p})\}$ and unlabeled target domain data $D_q^u := \{X_{n_p+1}, \ldots, X_{n_p+n_q}\}$. Based on these samples $D := (D_p, D_q^u)$, our goal is to find a classifier $\widehat{h}_q : \mathcal{X} \to \mathcal{Y}$ for the target domain. It is well-known that the classification error of a classifier $h : \mathcal{X} \to \mathcal{Y}$, i.e. the risk $Q(h(X) \neq Y)$, is minimzed by the naive Bayes classifier $h_q(x) := \arg\max_{y \in [K]} q(y|x)$. For this reason, we aim at finding an estimator $\widehat{q}(y|x)$ of $q(y|x)$ to get the plug-in classifier given by $\widehat{h}_q(x) := \arg\max_{y \in [K]} \widehat{q}(y|x)$. In order to evaluate the quality of an estimator $\widehat{q}(y|x)$, we employ the cross-entropy loss $\ell(y, \widehat{q}(\cdot|x)) := -\log \widehat{q}(y|x)$. Then, the corresponding risk is given by $\mathcal{R}_q(\widehat{q}(y|x)) := \int_{\mathcal{X} \times \mathcal{Y}} \ell(y, \widehat{q}(\cdot|x)) \, dq(x, y)$ and the minimal risk is defined as $\mathcal{R}_q^* := \min_{\widehat{q}(y|x)} \mathcal{R}_q(\widehat{q}(y|x))$. The conditional probability $q(y|x)$ is the optimal (or true) predictor which achieves the minimal risk with respect to $\ell$ and $Q$, i.e. $\mathcal{R}_q^* = \mathcal{R}_q(q(y|x)) =$

$-\mathbb{E}_{x \sim q} \sum_{k \in [K]} q(k|x) \log q(k|x)$. Moreover, we use the notation $a_n \lesssim b_n$ to denote that there exists a constant $c > 0$ such that $a_n \leq c\, b_n$, for all $n \in \mathbb{N}$.

## 2.2 LABEL SHIFT

In this paper, we consider the label shift setting (Saerens et al., 2002; Storkey, 2009).

**Assumption 2.1** (**Label shift**). Let $P$ and $Q$ be two joint probability distributions defined on $\mathcal{X} \times \mathcal{Y}$. Suppose that $P$ has non-trivial class probabilities, i.e. $p(y), q(y) \in (0, 1)$, $\forall y \in [K]$. We say that $P$ and $Q$ satisfy the label shift assumption if the class-conditional probabilities of $P$ and $Q$ are the same, whereas their class probabilities differ, i.e.

$$p(x|y) = q(x|y) \qquad \text{and} \qquad p(y) \neq q(y). \tag{1}$$

To better illustrate label shift, we provide a real-world example. Imagine that $x \in \mathcal{X}$ represents symptoms of a disease and that the label $y \in \mathcal{Y}$ indicates whether a person has been infected with the disease. Moreover, assume that the distributions $P$ and $Q$ defined on $\mathcal{X} \times \mathcal{Y}$ correspond to the joint distribution of symptoms and infections in different hospitals, e.g. in different locations, that adopt distinct prevention and control measures so that the disease prevalence differs, i.e. $p(y) \neq q(y)$. At the same time, it is reasonable to assume that the symptoms of the disease and the mechanism that symptoms caused by diseases are the same in both places, i.e. $p(x|y) = q(x|y)$. To make a diagnostic model based on data from one of the hospitals work in the other hospital it becomes therefore crucial to study label shift adaptation.

The goal of label shift adaptation is to estimate $q(y|x)$ in the target domain. Estimating $q(y|x)$ can be converted into estimating the class probability ratio $w^* := (w_y^*)_{y \in [K]}$ between $q(y)$ and $p(y)$, i.e.

$$w_y^* := q(y)/p(y), \qquad y \in [K]. \tag{2}$$

To be specific, given the label-shift assumption 2.1, $q(y|x)$ can be expressed in terms of the true predictor $p(y|x)$ in the source domain and the class probability ratio $w^*$:

$$q(y|x) = \frac{w_y^* p(y|x)}{\sum_{k=1}^K w_k^* p(k|x)}, \qquad y \in [K]. \tag{3}$$

For a detailed proof of Eq. (3) see Saerens et al. (2002); Alexandari et al. (2020) or Appendix A. Note that the predictor $p(y|x)$ can be learned by fitting a calibrated ReLU network with a softmax output layer on the labeled samples in the source domain. Thus, in order to estimate the optimal predictor $q(y|x)$ in the target domain, it suffices to estimate the probability ratio $w^*$.

In addition to label shift adaptation, the label shift assumption also exists in long-tailed learning (Zhu et al., 2023) and imbalanced semi-supervised learning (Kim et al., 2020; Wei et al., 2021). While these two problems often assume a balanced class probability for test data, certain methods still require the estimation of the class probabilities for pre-trained data (Zhu et al., 2023) or unlabeled training data (Kim et al., 2020) to enhance the quality of pseudo-labels of data. Consequently, studying label shift adaptation can contribute to address the challenges in these two problems.

## 3 CLASS PROBABILITY MATCHING FRAMEWORK

In this section, in order to estimate $w^*$, we introduce a novel matching framework named *class probability matching* that motivates the proposed algorithm for label shift adaptation in Section 4. Before we proceed, we first revisit the existing feature probability matching framework.

**Revisit of feature probability matching methods.** Many existing methods that aim at estimating the ratio $w^*$ are based on matching two probability distributions of the feature variable $X$. To be specific, under the label shift assumption, it is shown in Zhang et al. (2013); Garg et al. (2020) that the marginal probability $q(x)$ has the following representation

$$q(x) = \sum_{y=1}^K q(y)q(x|y) = \sum_{y=1}^K w_y^* p(y)q(x|y) = \sum_{y=1}^K w_y^* p(y)p(x|y). \tag{4}$$

In order to determine $w^*$, feature probability matching methods (Zhang et al., 2013; Guo et al., 2020) aim to find a weight vector $w := (w_k)_{k \in [K]}$ such that

$$q(x) = \sum_{y=1}^{K} w_y p(y) p(x|y) \tag{5}$$

holds. In other words, they get $w^*$ by matching $q(x)$ with a weighted combination of $p(x|y)$. Theoretically, Garg et al. (2020) prove that $w = w^*$ is the unique solution of Eq. (5) under the assumption of linear independence of the set of distributions $\{p(x|y) : y \in [K]\}$.

Unfortunately, existing methods that match probability distributions on $X$ suffer from computational inefficiency. For example, the method proposed in Zhang et al. (2013) suffers from the computational complexity of $O(n_p^3)$. Moreover, training GANs to implicitly learn the distribution of $X$ in Guo et al. (2020) is computationally extremely costly. Thus, in order to avoid the above problem, we develop a method that matches probability distributions of the label variable $Y$ to get $w^*$. It is easy to see that the class probability $p(y)$ is the only probability distribution of $Y$ that can be estimated from the available data. Consequently, we investigate the method of matching $p(y)$ to get $w^*$.

**A new class probability matching framework.** By the law of total probability, we get a representation of the class probability $p(y)$ as

$$p(y) = \int_{\mathcal{X}} p(x) p(y|x) \, dx = \int_{\mathcal{X}} \frac{p(x)}{q(x)} q(x) p(y|x) \, dx, \quad y \in [K], \tag{6}$$

i.e. the class probability $p(y)$ can be written as a weighted integration with the marginal probability ratios $p(x)/q(x)$ of $X$ as the weight. Moreover, we find that the marginal probability ratio $p(x)/q(x)$ and the class probability ratios $w_y^* = q(y)/p(y)$, $y \in [K]$, are related. More precisely, under the label shift assumption, the representation of $q(x)$ in Eq. (4) together with Bayes formula yields

$$\frac{p(x)}{q(x)} = \frac{p(x)}{\sum_{y=1}^{K} w_y^* p(y) p(x|y)} = \frac{1}{\sum_{y=1}^{K} w_y^* p(y|x)}. \tag{7}$$

From Eq. (7), we can see that under the label shift assumption, the marginal probability ratios $p(x)/q(x)$ of $X$ can be parameterized by a simple $K$-dimensional vector $w^* = (w_y^*)_{y \in [K]}$. By incorporating Eq. (7) into Eq. (6), we get

$$p(y) = \int_{\mathcal{X}} \frac{p(y|x)}{\sum_{k=1}^{K} w_k^* p(k|x)} q(x) \, dx = \mathbb{E}_{x \sim q} \left( \frac{p(y|x)}{\sum_{k \in [K]} w_k^* p(k|x)} \right), \quad y \in [K].$$

Therefore, a new strategy to obtain $w^*$ is to find a weight vector $w := (w_k)_{k \in [K]}$ satisfying

$$p(y) = \mathbb{E}_{x \sim q} \left( \frac{p(y|x)}{\sum_{k \in [K]} w_k p(k|x)} \right), \quad y \in [K]. \tag{8}$$

Note that in Eq. (8) we match two class probability distributions: the class probability $p(y)$ and a weighted expectation of $p(y|x)$ with respect to the target distribution $q(x)$. We therefore call the aforementioned strategy *class probability matching* (*CPM*).

**Equivalence of class and feature probability matching frameworks.** In contrast to matching two distributions of the $d$-dimensional feature vector $X$ in Eq. (5), we match two distributions of the one-dimensional label $Y$ variable in Eq. (8). Since the label $Y$ only takes $K$ discrete values, class probability matching only needs to match $K$ equations, thereby allowing for a reduction of the computational cost of CPM.

The following equivalence theorem provides the theoretical guarantee that the class probability matching framework on $Y$ and the feature probability matching framework on $X$ in Eq. (5) share the same weight solution under the label shift assumption.

**Theorem 3.1** (**Equivalence of Matching on $X$ and Matching on $Y$**). *Let Assumption 2.1 hold. Then the weight vector $w := (w_k)_{k \in [K]}$ satisfies the feature probability matching on $X$ in Eq. (5) if and only if it satisfies the class probability matching on $Y$ in Eq. (8).*

Through the equivalence between the two matching frameworks, it becomes evident that the class probability ratio $w = w^*$ is also the unique solution to Eq. (8) under mild assumptions. The class probability matching framework motivates an algorithm which will be introduced in Section 4.

## 4 CLASS PROBABILITY MATCHING ALGORITHM

In this section, based on our proposed framework named *class probability matching* in Eq. (8) from Section 3, we present the empirical version of CPM and integrate calibrated networks to obtain an algorithm named *Class Probability Matching with Calibrated Networks* (CPMCN) for label shift adaptation. The proposed algorithm consists of two parts: the initial phase is to estimate the class probability ratio, while the second step is to obtain the corresponding classifier for the target domain.

**Estimation of the Probability Ratio $w^*$.** On the one hand, based on the source domain data, the left-hand side of Eq. (8) can be estimated by

$$\widehat{p}(y) := \frac{1}{n_p} \sum_{i=1}^{n_p} \mathbf{1}\{Y_i = y\}, \qquad y \in [K], \tag{9}$$

where $\mathbf{1}\{Y_i = y\}$ denotes the indicator function which takes the value $1$ if $Y_i = y$ and the value $0$ otherwise.

On the other hand, with the aid of target domain samples $D_q^u$, the right-hand side of Eq. (8) can be approximated by $n_q^{-1} \sum_{X_i \in D_q^u} \left( p(y|X_i)/(\sum_{k \in [K]} w_k p(k|X_i)) \right)$. In order to estimate the class membership probabilities $p(k|X_i)$ of target domain data $X_i \in D_q^u$ in the source domain, we first fit a calibrated network $\widehat{p}(y|x)$ with source data $D_p$ as an estimator of the predictor $p(y|x)$. Note that we employ calibrated networks here because ordinary neural networks struggle to provide accurate estimates of class membership probabilities, even though they are good classifiers. We construct calibrated networks via bias-corrected temperature scaling (BCTS) (Alexandari et al., 2020). Specifically, let $z(x)$ be the vector of the original softmax logits of the sample $x$. BCTS introduces a temperature scale $T$ and a bias vector $b := (b_k)_{k \in [K]}$ to get the calibrated predictor $\widehat{p}(y|x) := \exp(z(x)/T + b_y)/(\sum_{k \in [K]} \exp(z(x)/T + b_k))$ for $y \in [K]$, where $T$ and $b$ are both optimized with respect to the negative log-likelihood on the validation data. Then by using the calibrated network $\widehat{p}(y|x)$ to predict the class membership probabilities $\widehat{p}(y|X_i)$ for all target domain samples $X_i \in D_q^u$, the right-hand side of Eq. (8) can be estimated by

$$\widehat{p}_q^w(y) := \frac{1}{n_q} \sum_{X_i \in D_q^u} \frac{\widehat{p}(y|X_i)}{\sum_{k=1}^K w_k \widehat{p}(k|X_i)}, \qquad y \in [K]. \tag{10}$$

Since the probability ratio $w^*$ is a solution of matching on $Y$ in Eq. (8), $w^*$ can be estimated by matching the estimates $\widehat{p}(y)$ in Eq. (9) and $\widehat{p}_q^w(y)$ in Eq. (10). In order to match $\widehat{p}(y)$ and $\widehat{p}_q^w(y)$, we have to find the solution $\widehat{w}$ to the following minimization problem

$$\widehat{w} := \arg\min_{w \in \mathbb{R}^K} \sum_{y=1}^K \left| \widehat{p}(y) - \widehat{p}_q^w(y) \right|^2, \tag{11}$$

which can be solved by the Broyden-Fletcher-Goldfarb-Shanno (BFGS) algorithm (Broyden, 1970), with a computational complexity of $O(n_q K^2)$ in each iteration. By definition of $\widehat{p}_q^w$ in Eq. (10), there holds $\sum_{y \in [K]} \widehat{w}_y \widehat{p}_q^{\widehat{w}}(y) = 1$. Therefore, the weight $\widehat{w}$ that matches $\widehat{p}(y)$ and $\widehat{p}_q^{\widehat{w}}(y)$ is required to satisfy $\sum_{y \in [K]} \widehat{w}_y \widehat{p}(y) = 1$. Thus, for the solution $\widehat{w}$ in Eq. (11), we take the normalized weight $\widehat{w}_k/(\sum_{y \in [K]} \widehat{w}_y \widehat{p}(y))$ as the final estimation $\widehat{w}_k$, for all $k \in [K]$.

**Obtaining a classifier for the target domain.** Using the weighted predictor representation in Eq. (3), the calibrated network predictor $\widehat{p}(y|x)$, and the estimate $\widehat{w}$ of the ratio $w^*$ in Eq. (11), we are able to estimate $q(y|x)$ in the target domain by

$$\widehat{q}(y|x) = \frac{\widehat{w}_y \widehat{p}(y|x)}{\sum_{j=1}^K \widehat{w}_j \widehat{p}(j|x)}. \tag{12}$$

Based on the estimator $\widehat{q}(y|x)$ in Eq. (12), we obtain the plug-in classifier

$$\widehat{h}_q(x) := \arg\max_{y \in [K]} \widehat{q}(y|x). \tag{13}$$

The above procedures for label shift adaptation are summarized in Algorithm 1.

**Comparison with feature probability matching methods.** CPMCN keeps the same theoretical guarantee for the weight solution as methods matching on $X$, but the computational complexity of CPMCN is much smaller: the computational complexity of Eq. (10) is $O(n_q K^2)$ per iteration, but methods matching on $X$ either require inversion of the Gram matrix which has a high computational complexity of $O(n_p^3)$ or is computationally very intensive due to implicitly learning the distribution of $X$ by a GAN model.

---

**Algorithm 1** Class Probability Matching with Calibrated Networks (CPMCN)

---

**Input:** Source domain samples $D_p := (X_i, Y_i)_{i=1}^{n_p}$;

Target domain samples $D_q^u := (X_i)_{i=n_p+1}^{n_p+n_q}$;

Compute the class probability $\widehat{p}(y)$ from the source domain by Eq. (9);

Fit the calibrated networks $\widehat{p}(y|x)$ with the source domain samples $D_p$, and make predictions $\widehat{p}(y|X_i)$ on target domain samples $X_i \in D_q^u$;

Compute the weighted combination of $\widehat{p}(y|X_i)$ to get $\widehat{p}_q^w(y)$ as in Eq. (10);

Solve the minimization problem in Eq. (11) to match $\widehat{p}(y)$ and $\widehat{p}_q^w(y)$ and obtain $\widehat{w}$;

Compute the predictor $\widehat{q}(y|x)$ by Eq. (12) and the induced classifier $\widehat{h}_q$ by Eq. (13).

**Output:** Predicted labels $\{\widehat{h}_q(X_i)\}_{i=n_p+1}^{n_p+n_q}$ of samples from the target domain.

---

**Comparison with prediction probability matching methods.** Matching methods on the prediction $\widehat{y}$ first learn a predictor $\widehat{h}$ in the source domain and then match the probability $q(\widehat{y})$ of the prediction $\widehat{y} = \widehat{h}(x)$. However, there exists an avoidable prediction error between the learned predictor $\widehat{h}$ and the true predictor $h$ such that matching $q(\widehat{y})$ is not an accurate approximation of matching the probability $q(h(x))$ of true prediction $h(x)$. By contrast, $p(y)$ can be easily estimated using the labels of source domain data, which do not involve such prediction error of $\widehat{h}$ and therefore matching $p(y)$ yields a more accurate estimate of $w^*$.

## 5 THEORETICAL RESULTS

In this section, we present the theoretical results of the proposed class probability matching framework in Section 3 and the CPMCN algorithm for label shift adaptation in Section 4. Firstly, we prove that the true ratio $w^*$ is the unique solution of the CPM framework formulated in Eq. (8), which justifies the use of the solution $\widehat{w}$ in the CPMCN algorithm to estimate $w^*$. Secondly, we establish an upper bound for the estimation error of $\widehat{w}$ and the generalization error bound for the classification performance of CPMCN, which aims to demonstrate the benefits of using calibrated networks in CPMCN.

Our theoretical results build upon the following assumption which is taken from Zhang et al. (2013); Garg et al. (2020).

**Assumption 5.1 (Linear Independence).** Assume that the class conditional probabilities $\{p(x|y)\}_{y\in[K]}$ are linearly independent.

**Remark 5.2.** *Assumption 5.1 is a weaker than Assumption A.3 in Lipton et al. (2018) which assumes that the confusion matrix whose $(i, j)$-th entry is $p(\widehat{y} = i, y = j)$ is invertible.*

The following theorem demonstrates existence and uniqueness of a solution to the equation system Eq. (8). In particular, it shows that the solution corresponds to the probability ratio $w^*$.

**Theorem 5.3 (Identifiability).** *Let Assumptions 2.1 and 5.1 hold. Moreover, let the probability ratio $w^*$ be as in Eq. (2). Then the equation system Eq. (8) holds if and only if $w = w^*$.*

Theorem 5.3 justifies the use of the solution $\widehat{w}$ to the minimization problem in Eq. (11) to estimate $w^*$, since Eq. (11) is the empirical version of the system of equations Eq. (8). The next theorem provides an upper bound on the estimation error of $\widehat{w}$ for estimating $w^*$.

**Theorem 5.4 (Weight Estimation Error Bound).** *Let Assumptions 2.1 and 5.1 hold. Moreover, denote the function set of calibrated networks as $\mathcal{F}$. Then, with probability at least $1 - 2/n_p - 2/n_q$, there holds*

$$\|\widehat{w} - w^*\|_2^2 \lesssim \big( \inf_{f\in\mathcal{F}} \mathcal{R}_p(f) - \mathcal{R}_p^* \big) + \sqrt{\mathrm{VC}(\mathcal{F}) \log n_p / n_p} + 2\sqrt{\log n_p / n_p} + 2\log n_q / n_q.$$

The upper bound of the estimation error of $\widehat{w}$ on the right-hand side of the above inequality consists of the upper bound of the *bias* term (first summand) and the *variance* term (last three summands).

The following discussion indicates that the utilization of calibrated neural networks in CPMCN significantly reduces the bias term while keeping the variance term almost unchanged, thus reducing the class probability ratio estimation error $\|\widehat{w} - w^*\|_2$.

**Reduction of the Bias Error.** The *bias* error term measures the approximation ability of the neural network function to the true $p(y|x)$. As stated in Alexandari et al. (2020), calibrated neural networks have significantly lower systematic bias and stronger approximation ability to $p(y|x)$. Specifically, Alexandari et al. (2020) proposed a calibrated method called BCTS, which learns bias and scaling parameters by optimizing the negative log-likelihood on a held-out training set to correct the probability output of neural networks $\widehat{p}(y|x)$. Therefore, the utilization of calibrated networks in CPMCN significantly reduces the bias error term of the estimation error of $\widehat{w}$.

**Negligible Increase in the Variance Error.** The *variance* error term is dominated by the term that includes the VC dimension of the function set $\mathcal{F}$ of calibrated neural networks. Bartlett et al. (2019) proved that the VC dimension of a neural network set depends on the number of weights $W$ and layers $L$. Compared with the commonly used networks, the calibrated networks used in the CPMCN algorithm have added only one more layer consisting of $K$ biases and $K$ scaling parameters. As $W$ and $L$ are usually much larger than $2K$ and $1$, respectively, the increase of the VC dimension calibrated networks compared to the original networks can be considered negligible. As a result, the increase in the variance error term is negligible, as well.

Finally, we present the generalization bound of our algorithm in the target domain.

**Theorem 5.5** (**Generalization Bound**). *Let Assumptions 2.1 and 5.1 hold. Moreover, let* $\mathrm{VC}(\mathcal{F})$ *be the VC dimension of the function set of calibrated neural networks* $\mathcal{F}$ *and let* $\widehat{q}(y|x)$ *be the estimator in Eq. (12). Then, with probability at least* $1 - 2/n_p - 2/n_q$, *there holds*

$$\mathcal{R}_q(\widehat{q}(y|x)) - \mathcal{R}_q^* \lesssim 3\left(\inf_{f \in \mathcal{F}} \mathcal{R}_p(f) - \mathcal{R}_p^*\right) + 3\sqrt{\mathrm{VC}(\mathcal{F})\log n_p / n_p} + 6\sqrt{\log n_p / n_p} + 6\log n_q / n_q.$$

Compared to the error bound of the class probability ratio estimation in Theorem 5.4, the only slight differences in the generalization error bound in Theorem 5.5 lie in the different constant coefficients in front of each term. As a result, similar discussions as above yield that the utilization of calibrated neural networks can improve the generalization performance of the CPMCN algorithm. All the proofs of the results presented in this section can be found in Appendix B.

# 6 EXPERIMENTS

In this section, we present experimental results on three benchmark datasets (MNIST LeCun et al. (2010), CIFAR10 Krizhevsky et al. (2009), and CIFAR100 Krizhevsky et al. (2009)) to validate the effectiveness of the proposed CPMCN algorithm in comparison to existing matching and EM-based methods on label shift adaptation tasks.

**Experimental Setups.** Based on the original benchmark datasets, the data for the label shift setting are constructed as follows. We take the training set of the benchmark datasets as the data for the source domain $D_p$ and reserve 10000 samples out of $D_p$ as a hold-out validation set, which is used to tune the hyper-parameters of the calibrated BCTS model (Alexandari et al., 2020). In order to generate the data for the target domain, $D_q$, we resample the original test set according to the label distribution $q(y)$. By removing the labels, we obtain the unlabeled target domain data $D_q^u$. For repeating experiments of each method, we use the code from Geifman & El-Yaniv (2017) to train ten different network models with different random seeds. For each model, we perform 10 trials, where each trial consists of a different sampling of the validation set and a different sampling of the label-shifted target domain data. The total number of repetitions is 100.

**Types of Label Shifts.** We consider the following two label shift cases. For *Dirichlet shift*, we generate the class probability $q(y)$ by a Dirichlet distribution with parameters $(\alpha_j)_{j=1}^K$ satisfying $\alpha_j = \alpha$ for the first ten classes, i.e. for $j \in [10]$. The class probabilities of the remaining classes are set to 0. The label shift is more severe with a smaller $\alpha$. For *tweak-one shift*, we set the class probability of the fourth class as a parameter $\rho$ and the probabilities $q(y)$ of the remaining nine classes among the first ten classes are set to $(1 - \rho)/9$. Meanwhile, the class probabilities of all remaining classes are set to 0. We explore the Dirichlet shift with $\alpha \in \{0.1, 0.2, 0.5, 0.8, 1.0, 10\}$ and the tweak-one shift with $\rho \in \{0.01, 0.02, 0.05, 0.6, 0.8, 0.9\}$.

**Comparison Methods.** We compare the CPMCN method with the following algorithms for label shift adaptation: the matching methods LTF (Guo et al., 2020), (Zhang et al., 2013), BBSL (Lipton et al., 2018), RLLS (Azizzadenesheli et al., 2019), ELSA (Tian et al., 2023) and the maximum

likelihood method EM (Alexandari et al., 2020). The LTF method is a feature probability matching method on $X$, rendering the calibrated networks inapplicable. For the remaining comparison algorithms, we employ the BCTS method for calibration. For all methods, the calibrated predictor $\widehat{p}(y|x)$ is reweighted using the ratio estimation $\widehat{w}$ to determine the predictor $\widehat{q}(y|x)$ in the target domain via Eq. (12) and plug-in classifier by Eq. (13). Details on the implementation of all algorithms, including CPMCN, can be found in Appendix C.2.

**Evaluation Metrics.** We assess the efficacy of the algorithms by addressing the label shift adaptation problem from two perspectives. On the one hand, to measure the estimation error of the class probability ratio $w^*$, we employ two different metrics, MSE_EVEN and MSE_PROP, which means the original and weighted mean squared error between $w^*$ and its estimate $\widehat{w}$, respectively. On the other hand, the classification accuracy (ACC) in the target domain is used to directly assess the algorithms' performances. Note that better performance is indicated by smaller MSE_PROP, MSE_EVEN, and larger ACC. Detailed information regarding the three metrics can be found in Appendix C.1.

| $\alpha$ | 0.1 | | | 0.2 | | | 0.5 | | |
|---|---|---|---|---|---|---|---|---|---|
| Metrics | MSE_EVEN | MSE_PROP | ACC (%) | MSE_EVEN | MSE_PROP | ACC (%) | MSE_EVEN | MSE_PROP | ACC (%) |
| LTF | 5.91e+1 | 3.84e+3 | 65.30 | 7.73e+1 | 2.07e+3 | 80.15 | 2.71e+1 | 9.67e+2 | 80.35 |
| | (1.05e+1) | (1.29e+3) | (7.94) | (1.20e+1) | (1.03e+3) | (1.63) | (8.01e+0) | (3.82e+2) | (2.86) |
| BBSL | 1.70e+0 | 5.02e+1 | 93.03 | 1.26e+0 | 2.64e+1 | 91.53 | 8.38e-1 | 1.11e+1 | 89.98 |
| | (7.25e-1) | (3.91e+1) | (0.41) | (8.54e+0) | (2.94e+1) | (0.40) | (1.02e+1) | (1.54e+1) | (0.36) |
| RLLS | 1.12e+0 | 4.94e+1 | 93.30 | 9.17e-1 | 2.37e+1 | 92.15 | 6.35e-1 | 1.08e+1 | 90.50 |
| | (1.75e+0) | (1.47e+2) | (0.66) | (1.40e+0) | (1.17e+2) | (0.55) | (9.06e-1) | (6.36e+1) | (0.42) |
| ELSA | 5.43e+0 | 6.60e+1 | 93.88 | 4.24e+0 | 4.33e+1 | 92.58 | 2.76e+0 | 1.62e+1 | 90.88 |
| | (1.08e+1) | (1.14e+2) | (0.53) | (9.90e+0) | (8.28e+1) | (0.56) | (7.25e+0) | (3.87e+1) | (0.50) |
| EM | 2.31e-1 | 7.95e+0 | 94.58 | 1.82e-1 | 4.15e+0 | 93.18 | 1.32e-1 | 2.11e+0 | 91.30 |
| | (8.31e-2) | (4.87e+0) | (0.33) | (5.31e-2) | (2.49e+0) | (0.29) | (2.46e-2) | (9.80e-1) | (0.25) |
| CPMCN | **1.09e-1** | **3.61e+0** | **95.48** | **1.31e-1** | **2.30e+0** | **94.08** | **1.02e-1** | **1.53e+0** | **92.00** |
| | **(7.91e-2)** | **(4.74e+0)** | **(0.33)** | **(5.07e-2)** | **(2.19e+0)** | **(0.28)** | **(2.37e-2)** | **(9.55e-1)** | **(0.25)** |
| $\alpha$ | 0.8 | | | 1.0 | | | 10 | | |
| Metrics | MSE_EVEN | MSE_PROP | ACC (%) | MSE_EVEN | MSE_PROP | ACC (%) | MSE_EVEN | MSE_PROP | ACC (%) |
| LTF | 1.13e+1 | 2.18e+2 | 84.80 | 1.20e+1 | 4.64e+2 | 86.45 | 5.95e+0 | 7.42e+1 | 83.02 |
| | (3.30e+0) | (2.13e+2) | (1.14) | (4.76e+0) | (1.61e+2) | (1.55) | (3.29e+0) | (3.21e+1) | (5.86) |
| BBSL | 8.48e-1 | 1.07e+1 | 89.13 | 9.39e-1 | 1.11e+1 | 88.72 | 6.19e-1 | 4.37e+0 | 87.45 |
| | (9.67e+0) | (1.49e+1) | (0.35) | (8.20e+1) | (7.10e+0) | (0.36) | (1.42e+2) | (7.05e+0) | (0.31) |
| RLLS | 5.40e-1 | 9.43e+0 | 89.70 | 5.72e-1 | 8.31e+0 | 89.52 | 4.99e-1 | 4.41e+0 | 87.50 |
| | (5.73e-1) | (3.28e+1) | (0.35) | (8.52e-1) | (3.25e+1) | (0.38) | (6.69e-2) | (8.97e-1) | (0.14) |
| ELSA | 2.87e+0 | 1.88e+1 | 89.43 | 2.77e+0 | 1.32e+1 | 89.85 | 1.97e+0 | 7.55e+0 | 87.98 |
| | (4.61e+0) | (2.49e+1) | (0.37) | (2.84e+0) | (1.15e+1) | (0.41) | (2.34e+1) | (2.13e+0) | (0.43) |
| EM | 1.42e-1 | 1.55e+0 | 90.73 | 1.51e-1 | 1.55e+0 | 90.40 | 1.53e-1 | 1.35e+0 | 88.85 |
| | (2.06e-2) | (7.21e-1) | (0.22) | (3.06e-2) | (9.69e-1) | (0.26) | (9.63e-3) | (1.15e-1) | (0.13) |
| CPMCN | **1.27e-1** | **1.43e+0** | **91.28** | **1.32e-1** | **1.35e+0** | **90.80** | **1.44e-1** | **1.32e+0** | **89.15** |
| | **(1.86e-2)** | **(6.67e-1)** | **(0.22)** | **(3.15e-2)** | **(1.03e+0)** | **(0.26)** | **(9.42e-3)** | **(1.13e-1)** | **(0.13)** |

Table 1: Performance on CIFAR100 under Dirichlet shift. For each method, the first line corresponds to the median of the considered metrics among 100 repeating experiments. The second line is the standard deviation of these 100 values. The best results are marked in **bold**.

**Experimental Results.** Table 1 and Table 2 in Appendix C.3 provide the results of label shift adaptation tasks on the CIFAR100 dataset of the above methods for two types of label shift scenarios and different label shift parameters $\alpha$ and $\rho$. Experimental results on the MNIST and CIFAR10 datasets are presented in Tables 3, 4, 5, and 6 in Appendix C.3. For all datasets, especially for the CIFAR100 dataset, Table 1, shows that the proposed CPMCN method has the lowest estimation error with respect to the class probability ratio and the highest classification accuracy for different values of $\alpha$, which shows that matching on $Y$ outperforms the compared methods.

**Training Curve.** Figure 1 presents the training curve of our CPMCN algorithm and the corresponding curve of MSE of class probability ratio estimation on CIFAR100 under the tweak-one shift with $\rho = 0.01$. Both curves exhibit the same trend: as the number of iterations increases, the objective function and the ratio estimation error decrease to almost zero. This empirical result validates the effectiveness of the matching method on $Y$ for estimating the ratio $w^*$. Furthermore, beyond a certain number of iterations, both criteria maintain their near-zero status.

**The effect of the source domain predictor $\widehat{p}(y|x)$ on the performance.** We investigate the impact of the quality of the network predictor $\widehat{p}(y|x)$ trained in the source domain on the performance of $\widehat{q}(y|x)$ in the target domain. For this, we vary the number of training epochs for the network in the source domain and based on $\widehat{p}(y|x)$ trained with different epochs, we compute the corresponding predictors $\widehat{q}(y|x)$ in the target domain. Figure 2 presents the classification accuracy and cross-entropy loss of $\widehat{p}(y|x)$ in the source domain and that of $\widehat{q}(y|x)$ in the target domain at different training epochs under tweak-one shift with $\rho = 0.01$ on CIFAR100. As can be seen from Figures 2,

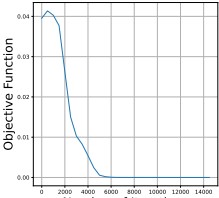 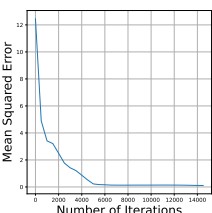 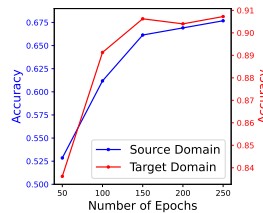 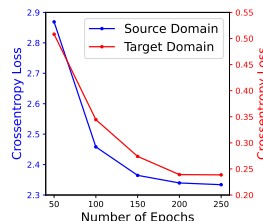

Figure 1: Training curve of CPMCN and ratio estimation error curve.

Figure 2: Effect of the source domain's network predictor on the performance in the target domain.

the better the network is trained in the source domain, the better the performance of CPMCN in the target domain. This validates the generalization bound of Theorem 5.5.

**Importance of Calibration in CPMCN.** To illustrate the role of calibration in CPMCN, we compare CPMCN without the calibrated model component (referred to as 'None') and CPMCN integrated with common calibration methods, including vector scaling (VS) (Guo et al., 2017), No Bias Vector Scaling (NBVS) (Alexandari et al., 2020), and BCTS. The comparison is conducted on the CIFAR100 dataset under the Dirichlet shift with parameter $\alpha = 10$. As shown in Figure 3, compared to the non-calibrated version, CPMCN combined with different calibration methods shows significant improvement both in ratio estimation and accuracy in the target domain, demonstrating the importance of calibration in CPMCN. This aligns with the theoretical explanation of the benefits from calibration in Section 5. Furthermore, besides the BCTS used in this paper, VS and NBVS also perform well.

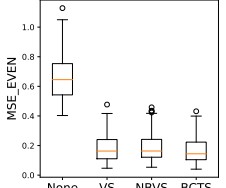 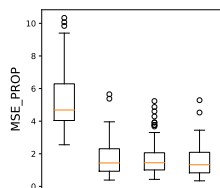 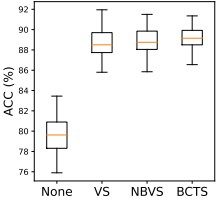 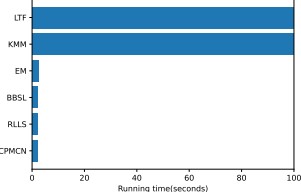

Figure 3: Boxplots for CPMCN using no calibration and various calibration techniques w.r.t. three different criteria. It is conducted under Dirichlet shift with $\alpha = 10$ on CIFAR100.

Figure 4: Running time on CIFAR100 under the Dirichlet shift with $\alpha = 0.1$.

**Less Running Time than Feature Probability Matching Methods.** Given that all compared methods use pre-trained network predictors, the time spent on pre-training is not included. We also include the running time of the KMM method from Zhang et al. (2013) in Figure 4, although it does not produce any results even after running for over 10,000 seconds. Figure 4 shows that compared with the feature probability matching methods, KMM and LTF, the class probability matching method CPMCN solved by L-BFGS-B (Zhu et al., 1997) has a significant computational advantage, which aligns with our complexity analysis. In addition, CPMCN's running time is comparable to and even faster than that of prediction matching methods including BBSL, RLLS, and EM-based methods by using the same optimizer.

## 7 CONCLUSION

To address the label shift adaptation problem, we propose a novel matching framework called class probability matching. CPM keeps the same theoretical guarantee as existing methods matching the distributions of the feature variable $X$, but is computationally more efficient due to directly matching the probability distribution of the label variable $Y$. Motivated by the CPM framework, we develop the CPMCN algorithm by utilizing calibrated neural networks for target domain classification. Theoretically, we establish CPMCN's generalization bound, highlighting the importance of incorporating calibrated networks. Experimentally, the CPMCN method performs better than other matching methods and EM-based algorithms. For future research, one research question would be to extend the method to the relaxed label shift setting in Garg et al. (2023); Tachet des Combes et al. (2020), where there is a concurrent shift in the class-conditional probability and the class probability.

ACKNOWLEDGMENTS

The authors would like to express the sincere gratitude to Tao Huang for his helpful discussions and suggestions throughout the experimental phase of this paper. His expertise and dedicated efforts significantly enhanced the quality and rigor of our experiments.

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

## A  PROOFS RELATED TO SECTION 2 AND 3

For the sake of completeness, we provide the proof of Eq. (3).

*Proof of Eq. (3).* Using Bayes' Formula, the label shift Assumption 2.1, and the law of total probability, we get

$$q(y|x) = \frac{q(x|y)q(y)}{\sum_{k=1}^{K} q(x|k)q(k)} = \frac{p(x|y)q(y)}{\sum_{k=1}^{K} p(x|k)q(k)}$$

$$= \frac{p(x|y)p(y)q(y)/p(y)}{\sum_{k=1}^{K} p(x|k)p(k)q(k)/p(k)} = \frac{p(x,y)w_y^*}{\sum_{k=1}^{K} p(x,k)w_k^*}$$

$$= \frac{(p(x,y)/p(x))w_y^*}{\sum_{k=1}^{K} (p(x,k)/p(x))w_k^*} = \frac{p(y|x)w_y^*}{\sum_{k=1}^{K} p(k|x)w_k^*}.$$

Here, the third equality holds since the nominator is first multiplied and then divided by $p(y)$ and and the terms in the denominator are first multiplied and then divided by $p(k)$. And the fifth equality holds since the nominator and denominator are divided by the same $p(x)$. This finishes the proof. ☐

*Proof of Theorem 3.1.* "Eq. (8) ⇒ Eq. (5)". Equation Eq. (8) together with Bayes' formula yields

$$p(y) = \mathbb{E}_{x \sim q} \frac{p(y|x)}{\sum_{j=1}^{K} w_j p(j|x)} = \mathbb{E}_{x \sim q} \frac{p(x|y)p(y)/p(x)}{\sum_{j=1}^{K} w_j p(x|j)p(j)/p(x)} = \mathbb{E}_{x \sim q} \frac{p(y)p(x|y)}{\sum_{j=1}^{K} w_j p(j)p(x|j)}.$$

By Assumption 2.1, we have $p(y) > 0$. Dividing both sides of the above equation by $p(y)$, we get

$$\mathbb{E}_{x \sim q} \frac{p(x|y)}{\sum_{j=1}^{K} w_j p(x|j)p(j)} = 1 \quad \Longleftrightarrow \quad \int_{\mathcal{X}} \frac{p(x|y)}{\sum_{j=1}^{K} w_j p(j)p(x|j)} \cdot q(x)\, dx = \int_{\mathcal{X}} p(x|y)\, dx. \tag{14}$$

Using the law of total probability and $p(x|k) = q(x|k)$ for all $k \in [K]$ by Assumption 2.1, we get

$$q(x) = \sum_{k=1}^{K} q(k)q(x|k) = \sum_{k=1}^{K} q(k)p(x|k). \tag{15}$$

Plugging Eq. (15) into Eq. (14), we obtain

$$\int_{\mathcal{X}} p(x|y) \cdot \frac{\sum_{k=1}^{K} q(k)p(x|k)}{\sum_{j=1}^{K} w_j p(j)p(x|j)}\, dx = \int_{\mathcal{X}} p(x|y)\, dx,$$

which is equivalent to

$$\int_{\mathcal{X}} p(x|y) \cdot \frac{\sum_{k=1}^{K} (q(k) - w_k p(k))p(x|k)}{\sum_{j=1}^{K} w_j p(j)p(x|j)}\, dx = 0. \tag{16}$$

Multiplying both sides of Eq. (16) by $(q(y) - w_y p(y))$ and taking the summation from $y = 1$ to $K$, we obtain

$$\int_{\mathcal{X}} \frac{\left( \sum_{k=1}^{K} (q(k) - w_k p(k))p(x|k) \right)^2}{\sum_{j=1}^{K} w_j p(j)p(x|j)}\, dx = 0.$$

By Assumption 2.1, we have $p(x|k) = q(x|k)$ and thus

$$\int_{\mathcal{X}} \frac{\left( \sum_{k=1}^{K} (q(k) - w_k p(k)) q(x|k) \right)^2}{\sum_{j=1}^{K} w_j p(j) q(x|j)} \, dx = 0.$$

Since $\sum_{j=1}^{K} w_j p(j) q(x|j) > 0$, there must hold $\sum_{k=1}^{K} (q(k) - w_k p(k)) q(x|k) = 0$ and therefore, we obtain $\sum_{k=1}^{K} q(k) q(x|k) = \sum_{k=1}^{K} w_k p(k) q(x|k)$. Again, using the law of total probability and Assumption 2.1, we get

$$q(x) = \sum_{k=1}^{K} q(k) q(x|k) = \sum_{k=1}^{K} w_k p(k) q(x|k) = \sum_{k=1}^{K} w_k p(k) p(x|k), \qquad x \in \mathcal{X}.$$

"Eq. (5) $\Rightarrow$ Eq. (8)". Using Bayes' formula and Eq. (5), we obtain

$$\mathbb{E}_{x \sim q} \frac{p(y|x)}{\sum_{j=1}^{K} w_j p(j|x)} = \mathbb{E}_{x \sim q} \frac{p(x|y) p(y)/p(x)}{\sum_{j=1}^{K} w_j p(x|j) p(j)/p(x)} = \mathbb{E}_{x \sim q} \frac{p(y) p(x|y)}{\sum_{j=1}^{K} w_j p(j) p(x|j)}$$

$$= \int_{\mathcal{X}} \frac{p(y) p(x|y)}{\sum_{j=1}^{K} w_j p(j) p(x|j)} \cdot q(x) \, dx = \int_{\mathcal{X}} \frac{p(y) p(x|y)}{\sum_{j=1}^{K} w_j p(j) p(x|j)} \cdot \left( \sum_{k=1}^{K} w_k p(k) p(x|k) \right) dx$$

$$= \int_{\mathcal{X}} p(y) p(x|y) \, dx = p(y), \qquad y \in [K],$$

which yields Eq. (8) and thus completes the proof. $\qquad\square$

## B   Proofs Related to Section 5

### B.1   Proof of Remark 5.2

*Proof of Remark 5.2.* Suppose that Assumption A.3 in Lipton et al. (2018) holds, i.e., the expected confusion matrix $\mathcal{C}_p(\widehat{h}) := p(\widehat{h}(x), y)$ is assumed to be invertible. By Assumption 2.1, we have $p(x|y) = q(x|y)$. Therefore, we only need to prove the linear independence of $\{p(x|y)\}_{y \in [K]}$. To this end, we assume that there exist some weights $(w_y)_{y \in [K]}$ such that $\sum_{y=1}^{K} w_y p(x|y) = 0$. Then for any $i \in [K]$, we have

$$\sum_{y=1}^{K} w_y p(\widehat{h}(X) = i|y) = \sum_{y=1}^{K} w_y \int_{\{\widehat{h}(x)=i\}} p(x|y) \, dx = \int_{\{\widehat{h}(x)=i\}} \sum_{y=1}^{K} w_y p(x|y) \, dx = 0.$$

By definition, we have $p(\widehat{h}(X) = i|y) = p(\widehat{h}(X) = i, y)/p(y)$. Therefore,

$$\sum_{y=1}^{K} w_y \cdot \frac{p(\widehat{h}(X) = i, y)}{p(y)} = 0, \qquad i \in [K].$$

With the notation $p(\widehat{h}(x), y) := (p(\widehat{h}(X) = i, y))_{i=1}^{K}$ the above $K$ equations can be reformulated as

$$\sum_{y=1}^{K} \frac{w_y}{p(y)} \cdot p(\widehat{h}(x), y) = 0.$$

Since the expected confusion matrix $\mathcal{C}_p(\widehat{h}) := p(\widehat{h}(x), y)$ is assumed to be invertible, the vectors $\{p(\widehat{h}(x), y)\}_{y \in [K]}$ are linear independent. Therefore, the coefficients $w_y/p(y)$ must be equal to zero, which implies $w_y$'s are all equal to zero. Thus, $\{p(x|y)\}_{y \in [K]}$ are linear independent and $\{q(x|y)\}_{y \in [K]}$ as well. This finishes the proof. $\qquad\square$

### B.2 Proof of Theorem 5.3

*Proof of Theorem 5.3.* "$\Rightarrow$" Suppose that the equation system Eq. (8) holds for some weight $w :=$ $(w_y)_{y\in[K]}$. The equivalence of Eq. (8) and Eq. (5) in Theorem 3.1 together with Assumption 2.1 yields

$$q(x) = \sum_{y=1}^{K} w_y p(y) p(x|y) = \sum_{y=1}^{K} w_y p(y) q(x|y).$$

By Assumption 5.1, we get $w_y p(y) = q(y)$ and thus $w_y = q(y)/p(y) = w_y^*$, $y \in [K]$, i.e., $w = w^*$. "$\Leftarrow$" Plugging $w = w^*$ into the right-hand side of Eq. (8), we obtain

$$\mathbb{E}_{x\sim q}\frac{p(y|x)}{\sum_{j=1}^{K} w_k^* p(j|x)} = \mathbb{E}_{x\sim q}\frac{p(x|y)p(y)/p(x)}{\sum_{j=1}^{K} w_j^* p(x|j)p(j)/p(x)} = \mathbb{E}_{x\sim q}\frac{p(y)p(x|y)}{\sum_{j=1}^{K} w_j^* p(j)p(x|j)}$$

$$= \int_{\mathcal{X}}\frac{p(y)p(x|y)}{\sum_{j=1}^{K} q(j)q(x|j)}\cdot q(x)\,dx = \int_{\mathcal{X}}\frac{p(y)p(x|y)}{q(x)}\cdot q(x)\,dx$$

$$= \int_{\mathcal{X}} p(y)p(x|y)\,dx = p(y),$$

which proves the conclusion. $\qquad\square$

### B.3 Proof of Theorem 5.4

Before we proceed, we provide some basic notations. For $1 \le p < \infty$, the $L_p$-norm of $x = (x_1,\dots,x_d)$ is defined as $\|x\|_p := (|x_1|^p + \dots + |x_d|^p)^{1/p}$, and the $L_\infty$-norm is defined as $\|x\|_\infty := \max_{i=1,\dots,d}|x_i|$. For any $a, b \in \mathbb{R}$, we denote $a \wedge b := \min\{a, b\}$ and $a \vee b := \max\{a, b\}$ as the smaller and larger value of $a$ and $b$, respectively. In addition, we define the network predictor $\widehat{p}(y|x)$ trained in the source domain as follows. Within the neural network space $\mathcal{F}$, the empirical risk minimizer on source domain data $D_p$ with respect to the cross-entropy loss is given by

$$\widehat{p}(y|x) := \arg\min_{f\in\mathcal{F}} \widehat{\mathcal{R}}_p(f) := \arg\min_{f\in\mathcal{F}} -\frac{1}{n_p}\sum_{i=1}^{n_p}\log(f_{Y_i}(X_i)), \tag{17}$$

where $f := (f_j)_{j=1}^{K}$ are the mappings from $\mathcal{X}$ to the $(K-1)$-dimensional simplex $\Delta^{K-1}$ with $f_j : \mathcal{X} \to [0,1]$, and $f_j(x)$ denoting the estimated probability of the label $j$ given the input $x$.

With these preparations, we provide the proof sketch of Theorem 5.4 as follows.

Step 1: Prove the identifiablity of $w^*$. In other words, $w^*$ is the unique solution of population CPM (Theorem 5.3).

Step 2: Note that $\widehat{w}$ is the solution of empirical CPM Eq. (11). Prove how the difference between $\widehat{w}$ and $w^*$ is affected by the excess risk of the predictor $\widehat{p}(y|x)$ in the source domain (Proposition B.1).

Step 3: Analyze the excess risk of the predictor $\widehat{p}(y|x)$ by considering its approximation error $\inf_{f\in\mathcal{F}}\mathcal{R}_p(f) - \mathcal{R}_p^*$ and sample error related to VC dimension of the network space $\mathcal{F}$ (Proposition B.6).

In Step 1 and 2, we use the label shift Assumption 2.1 and linear independence Assumption 5.1, which establishes linear independene of the class conditional probabilities $\{p(x|y)\}_{y\in[K]}$. We provide the following example to illustrate Assumption 5.1. If, e.g. $x$ encodes the symptoms of a disease and $y$ whether a person is infected with that same disease (0 indicating a healthy person and 1 an infected person), dependence of $\{p(x|y)\}_{y\in\{0,1\}}$ means that, for all $x$, $p(x|0) = \lambda p(x|1)$. From this it easily follows that $p(x|0) = p(x|1)$ and due to the label shift assumption $q(x|0) = q(x|1)$, i.e. it is impossible in source and target domain to distinguish healthy from infected persons through consideration of the alleged symptoms encoded in $x$.

To prove Theorem 5.4, we need the following Propositions B.1 and B.6. Proposition B.1 shows that the weight estimation error bound can be upper bounded with the aid of the excess risk of the predictor $\widehat{p}(y|x)$ in the source domain.

**Proposition B.1.** *Let Assumptions 2.1 and 5.1 hold. Moreover, let the predictor $\widehat{p}(y|x)$ be as in Eq. (17). Then with probability at least $1 - 1/n_p - 1/n_q$, there holds*

$$\|\widehat{w} - w^*\|_2^2 \lesssim \mathcal{R}_p(\widehat{p}(y|x)) - \mathcal{R}_p^* + \frac{\log n_q}{n_q} + \frac{\log n_p}{n_p}.$$

To prove Proposition B.1, we need the following Lemmas B.2, B.3, B.4, and B.5. Lemma B.2 was introduced in Bernstein (1946) and can be found in many statistical learning textbooks, see e.g., Massart (2007); Cucker & Zhou (2007); Steinwart & Christmann (2008).

**Lemma B.2** (Bernstein's inequality). *Let $B > 0$ and $\sigma > 0$ be real numbers, and $n \geq 1$ be an integer. Furthermore, let $\xi_1, \ldots, \xi_n$ be independent random variables satisfying $\mathbb{E}_P \xi_i = 0$, $\|\xi_i\|_\infty \leq B$, and $\mathbb{E}_P \xi_i^2 \leq \sigma^2$ for all $i = 1, \ldots, n$. Then for all $\tau > 0$, we have*

$$\mathrm{P}\left(\frac{1}{n}\sum_{i=1}^{n} \xi_i \geq \sqrt{\frac{2\sigma^2\tau}{n}} + \frac{2B\tau}{3n}\right) \leq e^{-\tau}.$$

**Lemma B.3.** *Let $\widehat{p}(y)$ be the estimate of the class probability in Eq. (9). Then with probability at least $1 - 1/n_p$, there holds*

$$\sum_{y=1}^{K} |\widehat{p}(y) - p(y)|^2 \lesssim \frac{\log n_p}{n_p}.$$

*Proof of Lemma B.3.* For $i \in [n]$ and $k \in [K]$, let the random variables $\xi_{i,k} := \mathbf{1}\{Y_i = k\} - p(k)$. Then we have $\mathbb{E}\xi_{i,k} = 0$, $\|\xi_{i,k}\|_\infty \leq 1$ and $\mathbb{E}\xi_{i,k}^2 = p(k)(1 - p(k)) \leq p(k)$. Applying Bernstein's inequality in Lemma B.2, we get

$$\left|\frac{1}{n_p}\sum_{i=1}^{n_p} \xi_{i,k}\right| = |\widehat{p}(k) - p(k)| \leq \sqrt{\frac{2p(k)\tau}{n_p}} + \frac{2\tau}{3n_p}$$

with probability at least $1 - 2e^{-\tau}$. Using the union bound and $(a + b)^2 \leq 2(a^2 + b^2)$, we obtain

$$\sum_{k=1}^{K} |\widehat{p}(k) - p(k)|^2 \leq \sum_{k=1}^{K} \left(\sqrt{\frac{2p(k)\tau}{n_p}} + \frac{2\tau}{3n_p}\right)^2 \leq \sum_{k=1}^{K} \left(\frac{4p(k)\tau}{n_p} + \frac{4\tau^2}{9n_p^2}\right)$$

with probability at least $1 - 2Ke^{-\tau}$. Taking $\tau := \log(2Kn_p)$, we get

$$\sum_{k=1}^{K} |\widehat{p}(k) - p(k)|^2 \leq \frac{4\log(2Kn_p)}{n_p} + \frac{4\log^2(2Kn_p)}{9n_p^2} \lesssim \frac{\log n_p}{n_p}$$

with probability at least $1 - 1/n_p$. This proves the assertion. $\qquad\square$

The following lemma shows that the $L_2$-distance between $\widehat{p}(y|x)$ and $p(y|x)$ can be upper bounded by the excess risk of $\widehat{p}(y|x)$ in the source domain.

**Lemma B.4.** *Let $\widehat{p}(y|x)$ be the estimator of $p(y|x)$. Then for any $y = 1, \ldots, K$, we have*

$$\int_{\mathcal{X}} (\widehat{p}(y|x) - p(y|x))^2 p(x)\, dx \leq \mathcal{R}_p(\widehat{p}(y|x)) - \mathcal{R}_p^*.$$

*Proof of Lemma B.4.* For any $k \in [K]$, there holds

$$\mathcal{R}_p(\widehat{p}(y|x)) - \mathcal{R}_p^* - \mathbb{E}_{x\sim p}|\widehat{p}(y|x) - p(y|x)|^2$$

$$= \mathbb{E}_{x\sim p} \sum_{j=1}^{K} -p(j|x) \log \frac{\widehat{p}(j|x)}{p(j|x)} - |\widehat{p}(k|x) - p(k|x)|^2. \qquad (18)$$

For any vector $u := (u_1, \ldots, u_K) \in (0, 1)^K$, we define

$$h_k(u) := -\sum_{j=1}^{K} p(j|x) \log \frac{u_j}{p(j|x)} - |u_k - p(k|x)|^2, \qquad k \in [K].$$

To find the minimum of $h_k(u)$ under the constraints $\sum_{k=1}^{K} u_k = 1$, we consider the Lagrange function

$$J_k(u) := h_k(u) + \alpha\left(\sum_{j=1}^{K} u_j - 1\right)$$

with the multiplier $\alpha > 0$. Taking the derivative w.r.t $u_j, j = 1, \dots, K$, and setting them to be zero, i.e.

$$\frac{\partial J_k(u)}{\partial u_k} = -\frac{p(k|x)}{u_k} - 2(u_k - p(k|x)) + \alpha = 0, \qquad k \in [K], \tag{19}$$

$$\frac{\partial J_k(u)}{\partial u_j} = -\frac{p(j|x)}{u_j} + \alpha = 0, \qquad j \neq k. \tag{20}$$

Since $\sum_{j=1}^{K} p(j|x) = 1$ and $\sum_{j=1}^{K} u_j = 1$, Eq. (20) yields

$$1 - p(k|x) = \sum_{j \neq k} p(j|x) = \sum_{j \neq k} \alpha u_j = \alpha(1 - u_k), \qquad k \in [K].$$

Therefore, we have $\alpha = (1 - p(k|x))/(1 - u_k)$. Plugging this into Eq. (19), we get

$$-\frac{p(k|x)}{u_k} - 2(u_k - p(k|x)) + \frac{1 - p(k|x)}{1 - u_k} = 0,$$

which is equivalent to

$$\frac{(u_k - p(k|x))(1 - 2u_k(1 - u_k))}{u_k(1 - u_k)} = 0$$

and thus $(u_k - p(k|x))(1 - 2u_k(1 - u_k)) = 0$. Since $u_k \in (0,1)$, we have $u_k(1 - u_k) \leq 1/4$, which implies $2u_k(1 - u_k) \leq 1/2 < 1$ and thus $1 - 2u_k(1 - u_k) > 0$. Therefore, there must hold $u_k - p(k|x) = 0$ for any $k \in [K]$. In other words, $\eta_P$ is the minimizer of $h_k$ for any $k \in [K]$. As a result, we have $h_k(u) \geq h_k(p(y|x)) = 0$ for any $u \in (0,1)^K$ satisfying $\sum_{j=1}^{K} u_j = 1$. By taking $u := \widehat{p}(y|x)$, we get

$$h_k(\widehat{p}(y|x)) - h_k(p(y|x)) = \sum_{j=1}^{K} -p(j|x) \log \frac{\widehat{p}(j|x)}{p(j|x)} - |\widehat{p}(k|x) - p(k|x)|^2 \geq 0.$$

This together with Eq. (18) yields the assertion. $\qquad\square$

**Lemma B.5.** *Let Assumption 2.1 hold. Then with probability at least $1 - 1/n_q$, we have*

$$\sum_{k=1}^{K} \left| \frac{1}{n_q} \sum_{i=n_p+1}^{n_p+n_q} \frac{\widehat{p}(k|X_i)}{\sum_{j=1}^{K} \widehat{w}_j \widehat{p}(j|X_i)} - \mathbb{E}_{x \sim q} \frac{p(k|x)}{\sum_{j=1}^{K} \widehat{w}_j p(j|x)} \right|^2 \lesssim \frac{\log n_q}{n_q} + \mathcal{R}_p(\widehat{p}(y|x)) - \mathcal{R}_p^*,$$

$$\sum_{k=1}^{K} \left| \frac{1}{n_q} \sum_{i=n_p+1}^{n_p+n_q} \frac{\widehat{P}(k|X_i)}{\sum_{j=1}^{K} w_j^* \widehat{p}(j|X_i)} - \mathbb{E}_{x \sim q} \frac{p(k|x)}{\sum_{j=1}^{K} w_j^* p(j|x)} \right|^2 \lesssim \frac{\log n_q}{n_q} + \mathcal{R}_p(\widehat{p}(y|x)) - \mathcal{R}_p^*.$$

*Proof of Lemma B.5.* First, we have

$$\left| \frac{\widehat{p}(k|x)}{\sum_{j=1}^{K} w_j \widehat{p}(j|x)} - \frac{p(k|x)}{\sum_{j=1}^{K} w_j p(j|x)} \right|$$

$$= \frac{|\widehat{p}(k|x) \sum_{j \neq k} w_j p(j|x) - p(k|x) \sum_{j \neq k} w_j \widehat{p}(j|x)|}{\left(\sum_{j=1}^{K} w_j \widehat{p}(j|x)\right)\left(\sum_{j=1}^{K} w_j p(j|x)\right)} \leq \frac{\sum_{j \neq k} w_j |p(j|x)\widehat{p}(k|x) - p(k|x)\widehat{p}(j|x)|}{\left(\sum_{j=1}^{K} w_j \widehat{p}(j|x)\right)\left(\sum_{j=1}^{K} w_j p(j|x)\right)}$$

$$\leq \frac{\sum_{j \neq k} w_j \left(|p(j|x)\widehat{p}(k|x) - p(j|x)p(k|x) + p(k|x)p(j|x) - p(k|x)\widehat{p}(j|x)|\right)}{\left(\sum_{j=1}^{K} w_j \widehat{p}(j|x)\right)\left(\sum_{j=1}^{K} w_j p(j|x)\right)}$$

$$= \frac{|\widehat{p}(k|x) - p(k|x)|}{\sum_{j=1}^{K} w_j \widehat{p}(j|x)} + \frac{p(k|x) \sum_{j \neq k} w_j |p(j|x) - \widehat{p}(j|x)|}{\left(\sum_{j=1}^{K} w_j \widehat{p}(j|x)\right)\left(\sum_{j=1}^{K} w_j p(j|x)\right)}. \tag{21}$$

Since for any $k \in [K]$, there exists a constant $\delta \in (0, 1/K)$ such that $p(k) \geq \delta$ and $q(k) \geq \delta$, we get $q(k) = 1 - \sum_{j \neq k} q(j) \geq 1 - (K-1)\delta$ and $p(k) = 1 - \sum_{j \neq k} p(j) \geq 1 - (K-1)\delta$. Therefore we have

$$\frac{\delta}{1 - (K-1)\delta} \leq w_k^* = \frac{q(k)}{p(k)} \leq \frac{1 - (K-1)\delta}{\delta}.$$

Therefore, we can set $w_k \in [c, 1/c]$ with a constant $c \in (0, \delta/(1 - (K-1)\delta)]$. Then for any $x$, we have $\sum_{j=1}^{K} w_j \widehat{p}(j|x) \geq c$ and $\sum_{j=1}^{K} w_j \widehat{p}(j|x) \geq c$. This together with the triangle inequality, Eq. (21), $a_1^2 + a_2^2 \leq (a_1 + a_2)^2$ and $\sum a_k^2 \leq (\sum a_k)^2$ for $a_k \geq 0$ yields

$$\sum_{j=1}^{K} \left| \int_{\mathcal{X}} \left( \frac{\widehat{p}(k|x)}{\sum_{j=1}^{K} w_j \widehat{p}(j|x)} - \frac{p(k|x)}{\sum_{j=1}^{K} w_j p(j|x)} \right) q(x)\, dx \right|^2$$

$$\leq \int_{\mathcal{X}} \sum_{j=1}^{K} \left| \frac{\widehat{p}(k|x)}{\sum_{j=1}^{K} w_j \widehat{p}(j|x)} - \frac{p(k|x)}{\sum_{j=1}^{K} w_j p(j|x)} \right|^2 q(x)\, dx$$

$$\leq \int_{\mathcal{X}} \sum_{k=1}^{K} \left( \frac{|\widehat{p}(k|x) - p(k|x)|}{\sum_{j=1}^{K} w_j \widehat{p}(j|x)} + \frac{p(k|x) \sum_{j \neq k} w_j |p(j|x) - \widehat{p}(j|x)|}{\left(\sum_{j=1}^{K} w_j \widehat{p}(j|x)\right)\left(\sum_{j=1}^{K} w_j p(j|x)\right)} \right)^2 q(x)\, dx$$

$$\leq 2 \int_{\mathcal{X}} \sum_{k=1}^{K} \left( \left( \frac{|\widehat{p}(k|x) - p(k|x)|}{\sum_{j=1}^{K} w_j \widehat{p}(j|x)} \right)^2 + \left( \frac{p(k|x) \sum_{j \neq k} w_j |p(j|x) - \widehat{p}(j|x)|}{\left(\sum_{j=1}^{K} w_j \widehat{p}(j|x)\right)\left(\sum_{j=1}^{K} w_j p(j|x)\right)} \right)^2 \right) q(x)\, dx$$

$$\leq \frac{2}{c^2} \sum_{k=1}^{K} \int_{\mathcal{X}} |\widehat{p}(k|x) - p(k|x)|^2 q(x)\, dx + \frac{2}{c^4} \int_{\mathcal{X}} \sum_{k=1}^{K} \left( p(k|x) \sum_{j \neq k} w_j |p(j|x) - \widehat{p}(j|x)| \right)^2 q(x)\, dx$$

$$\leq \frac{2}{c^2} \sum_{k=1}^{K} \int_{\mathcal{X}} |\widehat{p}(k|x) - p(k|x)|^2 q(x)\, dx + \frac{2}{c^4} \int_{\mathcal{X}} \left( \sum_{j=1}^{K} w_j |p(j|x) - \widehat{p}(j|x)| \right)^2 q(x)\, dx$$

$$\leq \frac{2}{c^2} \sum_{k=1}^{K} \int_{\mathcal{X}} |\widehat{p}(k|x) - p(k|x)|^2 q(x)\, dx + \frac{2}{c^6} \int_{\mathcal{X}} \left( \sum_{j=1}^{K} |p(j|x) - \widehat{p}(j|x)| \right)^2 q(x)\, dx$$

$$\leq \frac{2}{c^2} \sum_{k=1}^{K} \int_{\mathcal{X}} |\widehat{p}(k|x) - p(k|x)|^2 q(x)\, dx + \frac{2K}{c^6} \int_{\mathcal{X}} \left( \sum_{j=1}^{K} |p(j|x) - \widehat{p}(j|x)|^2 \right) q(x)\, dx$$

$$\lesssim \sum_{k=1}^{K} \int_{\mathcal{X}} |\widehat{p}(k|x) - p(k|x)|^2 q(x)\, dx. \tag{22}$$

Using the law of total probability, we get

$$\frac{q(x)}{p(x)} = \frac{\sum_{j=1}^{K} q(x|j)q(j)}{\sum_{j=1}^{K} p(x|j)p(j)} = \frac{\sum_{j=1}^{K} q(x|j)q(j)}{\sum_{j=1}^{K} q(x|j)p(j)} \leq \bigvee_{j=1}^{K} \frac{q(j)}{p(j)} \leq \frac{1 - (K-1)\delta}{\delta} \leq \frac{1}{c}.$$

This together with Eq. (22) and Cauchy-Schwarz Inequality yields

$$\sum_{j=1}^{K} \left| \int_{\mathcal{X}} \left( \frac{\widehat{p}(k|x)}{\sum_{j=1}^{K} w_j \widehat{p}(j|x)} - \frac{p(k|x)}{\sum_{j=1}^{K} w_j p(j|x)} \right) q(x)\, dx \right|^2$$

$$\leq \frac{2}{c} \sum_{k=1}^{K} \int_{\mathcal{X}} |\widehat{p}(k|x) - p(k|x)|^2 p(x)\, dx = \frac{2}{c} \sum_{k=1}^{K} \|\widehat{p}(k|\cdot) - p(k|\cdot)\|_{L_2(p(x))}^2$$

$$\lesssim \sum_{j=1}^{K} \|p(k|x) - \widehat{p}(k|x)\|_{L_2(p(x))}^2. \tag{23}$$

In addition, let

$$\xi_{i,k} := \frac{\widehat{p}(k|X_i)}{\sum_{j=1}^{K} w_j \widehat{p}(j|X_i)} - \mathbb{E}_{x \sim q} \frac{\widehat{p}(k|x)}{\sum_{j=1}^{K} w_j \widehat{p}(j|x)}.$$

Then we have $\mathbb{E}_{x \sim q} \xi_{i,k} = 0$,

$$\|\xi_{i,k}\|_\infty \leq \frac{1}{\sum_{j=1}^{K} w_j \widehat{p}(j|X_i)} + \mathbb{E}_{x \sim q} \frac{1}{\sum_{j=1}^{K} w_j \widehat{p}(j|x)}$$

$$\leq \left( \bigwedge_{j=1}^{K} w_j \sum_{j=1}^{K} \widehat{p}(j|X_i) \right)^{-1} + \left( \mathbb{E}_{x \sim q} \bigwedge_{j=1}^{K} w_j \sum_{j=1}^{K} \widehat{p}(j|x) \right)^{-1} \leq \frac{2}{c}$$

and

$$\mathbb{E}_{D_q^u} \xi_{i,k}^2 = \left( \mathbb{E}_{x \sim q} \frac{\widehat{p}(k|x)}{\sum_{j=1}^{K} w_j \widehat{p}(j|x)} \right) \left( 1 - \mathbb{E}_{x \sim q} \frac{\widehat{p}(k|x)}{\sum_{j=1}^{K} w_j \widehat{p}(j|x)} \right)$$

$$\leq \mathbb{E}_{x \sim q} \frac{\widehat{p}(k|x)}{\sum_{j=1}^{K} w_j \widehat{p}(j|x)} \leq \mathbb{E}_{x \sim q} \frac{1}{\sum_{j=1}^{K} w_j \widehat{p}(j|x)} = \mathbb{E}_{x \sim q} \left( \bigwedge_{j=1}^{K} w_j \sum_{j=1}^{K} \widehat{p}(j|x) \right)^{-1} \leq \frac{1}{c}.$$

Applying Bernstein's inequality in Lemma B.2 to $\xi_{i,k}$, $i \in [n_q]$, we get

$$\left| \frac{1}{n_q} \sum_{i=1}^{n_q} \xi_{i,k} \right| = \left| \frac{1}{n_q} \sum_{i=1}^{n_q} \frac{\widehat{p}(k|X_i)}{\sum_{j=1}^{K} w_j \widehat{p}(j|X_i)} - \mathbb{E}_{x \sim q} \frac{\widehat{p}(k|x)}{\sum_{j=1}^{K} w_j \widehat{p}(j|x)} \right| \leq \sqrt{\frac{2\tau c^{-1}}{n_q}} + \frac{4c^{-1}\tau}{3n_q}. \quad (24)$$

with probability at least $1 - 2e^{-\tau}$. Taking $\tau := 2 \log n_q$, we obtain

$$\frac{1}{n_q} \sum_{i=1}^{n_q} \frac{\widehat{p}(k|X_i)}{\sum_{j=1}^{K} w_j \widehat{p}(j|X_i)} - \mathbb{E}_{x \sim q} \frac{\widehat{p}(k|x)}{\sum_{j=1}^{K} w_j \widehat{p}(j|x)} \leq \sqrt{\frac{4c^{-1} \log n_q}{n_q}} + \frac{8c^{-1} \log n_q}{3n_q}$$

with probability at least $1 - 1/n_q$. Using $(a + b)^2 \leq 2(a^2 + b^2)$, Eq. (24) and Eq. (23), we obtain

$$\sum_{k=1}^{K} \left| \frac{1}{n_q} \sum_{i=n_p+1}^{n_p+n_q} \frac{\widehat{p}(k|X_i)}{\sum_{j=1}^{K} w_j \widehat{p}(j|X_i)} - \mathbb{E}_{x \sim q} \frac{p(k|x)}{\sum_{j=1}^{K} w_j p(j|x)} \right|^2$$

$$\leq 2 \sum_{k=1}^{K} \left| \frac{1}{n_q} \sum_{i=n_p+1}^{n_p+n_q} \frac{\widehat{p}(k|X_i)}{\sum_{j=1}^{K} w_j \widehat{p}(j|X_i)} - \mathbb{E}_{x \sim q} \frac{\widehat{p}(k|x)}{\sum_{j=1}^{K} w_j \widehat{p}(j|x)} \right|^2$$

$$+ 2 \sum_{k=1}^{K} \left| \mathbb{E}_{x \sim q} \frac{\widehat{p}(k|x)}{\sum_{j=1}^{K} w_j \widehat{p}(j|x)} - \mathbb{E}_{x \sim q} \frac{p(k|x)}{\sum_{j=1}^{K} w_j p(j|x)} \right|^2$$

$$\lesssim 2 \sum_{k=1}^{K} \left( \sqrt{\frac{4c^{-1} \log n_q}{n_q}} + \frac{8c^{-1} \log n_q}{3n_q} \right)^2 + 2 \sum_{k=1}^{K} \|p(k|x) - \widehat{p}(k|x)\|_{L_2(p(x))}^2$$

$$\lesssim \frac{\log n_q}{n_q} + \mathcal{R}_p(\widehat{p}(y|x)) - \mathcal{R}_p^*,$$

where the last inequality is due to Lemma B.4. Applying this result to $w := \widehat{w}$ and $w := w^*$, we obtain the assertion. $\qquad\square$

With the above lemmas, we are able to prove Proposition B.1.

*Proof of Proposition B.1.* Using $(a + b + c)^2 \leq 3(a^2 + b^2 + c^2)$, Lemma B.5 and B.3, we get

$$\sum_{k=1}^{K} \left| \mathbb{E}_{x \sim q} \frac{p(k|x)}{\sum_{j=1}^{K} \widehat{w}_j p(j|x)} - p(k) \right|^2$$

$$\leq 3 \sum_{k=1}^{K} \left| \mathbb{E}_{x \sim q} \frac{p(k|x)}{\sum_{j=1}^{K} \widehat{w}_j p(j|x)} - \frac{1}{n_q} \sum_{i=n_p+1}^{n_p+n_q} \frac{\widehat{p}(k|X_i)}{\sum_{j=1}^{K} \widehat{w}_j \widehat{p}(j|X_i)} \right|^2$$

$$+ 3 \sum_{k=1}^{K} \left| \frac{1}{n_q} \sum_{i=n_p+1}^{n_p+n_q} \frac{\widehat{p}(k|X_i)}{\sum_{j=1}^{K} \widehat{w}_j \widehat{p}(j|X_i)} - \widehat{p}(k) \right|^2 + 3 \sum_{k=1}^{K} \left| \widehat{p}(k) - p(k) \right|^2$$

$$\lesssim \frac{\log n_q}{n_q} + \mathcal{R}_p(\widehat{p}(y|x)) - \mathcal{R}_p^* + \sum_{k=1}^{K} \left| \frac{1}{n_q} \sum_{i=n_p+1}^{n_p+n_q} \frac{\widehat{p}(k|X_i)}{\sum_{j=1}^{K} \widehat{w}_j \widehat{p}(j|X_i)} - \widehat{p}(k) \right|^2 + \frac{\log n_p}{n_p} \quad (25)$$

with probability at least $1 - 1/n_p - 1/n_q$. By Eq. (11), we get the optimality of $\widehat{w}$ and thus

$$\sum_{k=1}^{K} \left| \frac{1}{n_q} \sum_{i=n_p+1}^{n_p+n_q} \frac{\widehat{p}(k|X_i)}{\sum_{j=1}^{K} \widehat{w}_j \widehat{p}(j|X_i)} - \widehat{p}(k) \right|^2 \leq \sum_{k=1}^{K} \left| \frac{1}{n_q} \sum_{i=n_p+1}^{n_p+n_q} \frac{\widehat{p}(k|X_i)}{\sum_{j=1}^{K} w_j^* \widehat{p}(j|X_i)} - \widehat{p}(k) \right|^2.$$

This together with $(a + b + c)^2 \leq 3(a^2 + b^2 + c^2)$ yields

$$\sum_{k=1}^{K} \left| \frac{1}{n_q} \sum_{i=n_p+1}^{n_p+n_q} \frac{\widehat{p}(k|X_i)}{\sum_{j=1}^{K} \widehat{w}_j \widehat{p}(j|X_i)} - \widehat{p}(k) \right|^2 \leq \sum_{k=1}^{K} \left| \frac{1}{n_q} \sum_{i=n_p+1}^{n_p+n_q} \frac{\widehat{p}(k|X_i)}{\sum_{j=1}^{K} w_j^* \widehat{p}(j|X_i)} - \widehat{p}(k) \right|^2$$

$$\leq 3 \sum_{k=1}^{K} \left| \frac{1}{n_q} \sum_{i=n_p+1}^{n_p+n_q} \frac{\widehat{p}(k|X_i)}{\sum_{j=1}^{K} w_j^* \widehat{p}(j|X_i)} - \mathbb{E}_{x \sim q} \frac{p(k|x)}{\sum_{j=1}^{K} w_j^* p(j|x)} \right|^2$$

$$+ 3 \sum_{k=1}^{K} \left| \mathbb{E}_{x \sim q} \frac{p(k|x)}{\sum_{j=1}^{K} w_j^* p(j|x)} - p(k) \right|^2 + 3 \sum_{k=1}^{K} \left| p(k) - \widehat{p}(k) \right|^2$$

$$\lesssim \frac{\log n_q}{n_q} + \mathcal{R}_p(\widehat{p}(y|x)) - \mathcal{R}_p^* + \frac{\log n_p}{n_p}, \quad (26)$$

where the last inequality is due to Lemma B.5 and Theorem 5.3. Combining Eq. (25) and Eq. (26), we obtain

$$\sum_{k=1}^{K} \left| \mathbb{E}_{x \sim q} \frac{p(k|x)}{\sum_{j=1}^{K} \widehat{w}_j p(j|x)} - p(k) \right|^2 \lesssim \frac{\log n_q}{n_q} + \mathcal{R}_p(\widehat{p}(y|x)) - \mathcal{R}_p^* + \frac{\log n_p}{n_p} \quad (27)$$

with probability at least $1 - 1/n_p - 1/n_q$. Using Bayes' Formula, the law of total probability and Assumption 2.1, we get

$$\mathbb{E}_{x \sim q} \frac{p(y|x)}{\sum_{j=1}^{K} \widehat{w}_j p(j|x)} = \mathbb{E}_{x \sim q} \frac{p(x|y)p(y)/p(x)}{\sum_{j=1}^{K} \widehat{w}_j p(x|j)p(j)/p(x)} = \mathbb{E}_{x \sim q} \frac{p(y)p(x|y)}{\sum_{j=1}^{K} \widehat{w}_j p(j)p(x|j)}$$

$$= p(y) \int_{\mathcal{X}} \frac{p(x|y)}{\sum_{j=1}^{K} \widehat{w}_j p(j)p(x|j)} q(x) \, dx = p(y) \int_{\mathcal{X}} p(x|y) \frac{\sum_{j=1}^{K} q(j)q(x|j)}{\sum_{j=1}^{K} \widehat{w}_j p(j)p(x|j)} \, dx$$

$$= p(y) \int_{\mathcal{X}} q(x|y) \frac{\sum_{j=1}^{K} q(j)q(x|j)}{\sum_{j=1}^{K} \widehat{w}_j p(j)q(x|j)} \, dx = p(y) + p(y) \int_{\mathcal{X}} q(x|y) \frac{\sum_{j=1}^{K} (q(j) - \widehat{w}_j p(j))q(x|j)}{\sum_{j=1}^{K} \widehat{w}_j p(j)q(x|j)} \, dx$$

$$= p(y) + p(y) \sum_{j=1}^{K} (q(j) - \widehat{w}_j p(j)) \int_{\mathcal{X}} \frac{q(x|y)q(x|j)}{\sum_{j=1}^{K} \widehat{w}_j p(j)q(x|j)} \, dx$$

$$= p(y) + \sum_{j=1}^{K} (w_j^* - \widehat{w}_j) \int_{\mathcal{X}} \frac{p(j)p(y)q(x|j)q(x|y)}{\sum_{j=1}^{K} \widehat{w}_j p(j)q(x|j)} \, dx.$$

Denote the matrix $\mathcal{C} := (c_{yj})_{y,j \in [K]}$ with

$$c_{yj} := \int_{\mathcal{X}} \frac{p(j)p(y)q(x|j)q(x|y)}{\sum_{k=1}^{K} \widehat{w}_k p(k)q(x|k)} \, dx \qquad j = 1, \ldots, K.$$

Then we have

$$\mathbb{E}_{x \sim q} \frac{p(y|x)}{\sum_{k=1}^{K} \widehat{w}_k p(k|x)} - p(y) = \sum_{j=1}^{K} c_{yj}(w_j^* - \widehat{w}_j). \tag{28}$$

Now we prove that $\mathcal{C}$ is invertible by showing that the rows of $\mathcal{C}$ are linear independent. Assume that there exist $\alpha_1, \ldots, \alpha_K \in \mathbb{R}$ such that $\sum_{y=1}^{K} \alpha_y c_{yj} = 0$, $j \in [K]$, which implies

$$0 = \sum_{y=1}^{K} \alpha_y c_{yj} = \sum_{y=1}^{K} \alpha_y \int_{\mathcal{X}} \frac{p(j)p(y)q(x|j)q(x|y)}{\sum_{j=1}^{K} w_j p(j) q(x|j)} \, dx = \int_{\mathcal{X}} \frac{\sum_{y=1}^{K} \alpha_y p(y) q(x|y)}{\sum_{k=1}^{K} w_k p(k) q(x|k)} \cdot p(j) q(x|j) \, dx.$$

Multiplying the above equation by $\alpha_j$ and taking summation from $j = 1$ to $K$, we get

$$0 = \sum_{j=1}^{K} \alpha_j \int_{\mathcal{X}} \frac{\sum_{y=1}^{K} \alpha_y p(y) q(x|y)}{\sum_{k=1}^{K} w_k p(k) q(x|k)} \cdot p(j) q(x|j) \, dx = \int_{\mathcal{X}} \frac{\left(\sum_{y=1}^{K} \alpha_y p(y) q(x|y)\right)^2}{\sum_{k=1}^{K} w_k p(k) q(x|k)} \, dx.$$

This implies that $\sum_{y=1}^{K} \alpha_y p(y) q(x|y) = 0$. By Assumption 5.1, we get $\alpha_y p(y) = 0$ and therefore $\alpha_y = 0$. This proves that $\mathcal{C}$ is invertible. Denote the inverse matrix of $\mathcal{C}$ as $\mathcal{M} = (m_{ij})_{K \times K}$. Then Eq. (28) implies

$$|w_k^* - \widehat{w}_k| = \left| \sum_{y=1}^{K} m_{ky} \left( \mathbb{E}_{x \sim q} \frac{p(y|x)}{\sum_{j=1}^{K} \widehat{w}_j p(j|x)} - p(y) \right) \right| \lesssim \sum_{y=1}^{K} \left| \mathbb{E}_{x \sim q} \frac{p(y|x)}{\sum_{j=1}^{K} \widehat{w}_j p(j|x)} - p(y) \right|.$$

This together with $(\sum_{k=1}^{K} a_k)^2 \leq K \sum_{k=1}^{K} a_k^2$ and Eq. (27) yields

$$\sum_{k=1}^{K} |w_k^* - \widehat{w}_k|^2 \lesssim \sum_{k=1}^{K} \left( \sum_{y=1}^{K} \left| \mathbb{E}_{x \sim q} \frac{p(y|x)}{\sum_{j=1}^{K} \widehat{w}_j p(j|x)} - p(y) \right| \right)^2$$

$$= K \left( \sum_{y=1}^{K} \left| \mathbb{E}_{x \sim q} \frac{p(y|x)}{\sum_{j=1}^{K} \widehat{w}_j p(j|x)} - p(y) \right| \right)^2 \lesssim \left( \sum_{y=1}^{K} \left| \mathbb{E}_{x \sim q} \frac{p(y|x)}{\sum_{j=1}^{K} \widehat{w}_j p(j|x)} - p(y) \right| \right)^2$$

$$\lesssim K \sum_{y=1}^{K} \left| \mathbb{E}_{x \sim q} \frac{p(y|x)}{\sum_{j=1}^{K} \widehat{w}_j p(j|x)} - p(y) \right|^2 \lesssim \sum_{y=1}^{K} \left| \mathbb{E}_{x \sim q} \frac{p(y|x)}{\sum_{j=1}^{K} \widehat{w}_j p(j|x)} - p(y) \right|^2$$

$$\lesssim \mathcal{R}_p(\widehat{p}(y|x)) - \mathcal{R}_p^* + \frac{\log n_q}{n_q} + \frac{\log n_p}{n_p}$$

with probability at least $1 - 1/n_p - 1/n_q$. This finishes the proof. $\square$

**Proposition B.6.** *Let $\widehat{p}(y|x)$ be as in Eq. (17) and $\mathrm{VC}(\mathcal{F})$ be the VC dimension of the function set $\mathcal{F}$. Then with probability at least $1 - 1/n_p$, there holds*

$$\mathcal{R}_p(\widehat{p}(y|x)) - \mathcal{R}_p^* \lesssim \inf_{f \in \mathcal{F}} \mathcal{R}_p(f) - \mathcal{R}_p^* + \sqrt{\frac{\mathrm{VC}(\mathcal{F}) \log n_p}{n_p}} + \sqrt{\frac{\log n_p}{n_p}} + \frac{\log n_q}{n_q}.$$

*Proof of Proposition B.6.* By the triangle inequality and the optimality of $\widehat{p}(y|x)$, i.e. $\widehat{\mathcal{R}}_p(\widehat{p}(y|x)) \leq \widehat{\mathcal{R}}_p(f_0)$ where $f_0 \in \mathcal{F}$ satisfy $\mathcal{R}_p(f_0) := \inf_{f \in \mathcal{F}} \mathcal{R}_p(f)$, we obtain

$$\mathcal{R}_p(\widehat{p}(y|x)) - \mathcal{R}_p^* = \left( \mathcal{R}_p(\widehat{p}(y|x)) - \widehat{\mathcal{R}}_p(\widehat{p}(y|x)) \right) + \left( \widehat{\mathcal{R}}_p(\widehat{p}(y|x)) - \widehat{\mathcal{R}}_p(f_0) \right)$$

$$+ \left( \widehat{\mathcal{R}}_p(f_0) - \mathcal{R}_p(f_0) \right) + \left( \mathcal{R}_p(f_0) - \mathcal{R}_p^* \right)$$

$$\leq 2 \sup_{f \in \mathcal{F}} (\mathcal{R}_p(f) - \widehat{\mathcal{R}}_p(f)) + (\mathcal{R}_p(f_0) - \mathcal{R}_p^*)$$

$$= 2 \sup_{f \in \mathcal{F}} (\mathcal{R}_p(f) - \widehat{\mathcal{R}}_p(f)) + \inf_{f \in \mathcal{F}} \mathcal{R}_p(f) - \mathcal{R}_p^*.$$

Let the calibrated neural networks $f \in \mathcal{F}$ satisfy $f_k(x) \geq \delta$ with the constant $\delta \in (0, 1/K)$ for any $x$ and $y \in [K]$ such that $\ell(y, f(x)) = -\log f_y(x)$ is bounded. If we replace one sample $(X_i, Y_i)$ with another sample $(X_i', Y_i')$, the change in $\sup_{f \in \mathcal{F}}(\mathcal{R}_p(f) - \widehat{\mathcal{R}}_p(f))$ is not larger than $-\log \delta - (-\log(1 - \delta)) = \log((1 - \delta)/\delta)$. By applying McDiarmid's inequality McDiarmid (1989), we get with probability at least $1 - 1/n_p$,

$$\sup_{f \in \mathcal{F}} \mathcal{R}_p(f) - \widehat{\mathcal{R}}_p(f) \leq \mathbb{E}_{p^n} \sup_{f \in \mathcal{F}} \mathcal{R}_p(f) - \widehat{\mathcal{R}}_p(f) + \log \frac{1 - \delta}{\delta} \sqrt{\frac{\log n_p}{2n_p}}. \tag{29}$$

Define the function set $\mathcal{G} := \{g_f(x, y) = -\log(f_y(x)) : f \in \mathcal{F}\}$ and its expected Rademacher complexity

$$\mathrm{Rad}(\mathcal{G}, n) := \mathbb{E}_{(x_i, y_i) \sim p} \mathbb{E}_{\xi_i} \sup_{f \in \mathcal{F}} \frac{1}{n} \sum_{i=1}^{n} \xi_i g_f(x_i, y_i).$$

By applying the symmetrization in Proposition 7.10 of Steinwart & Christmann (2008), we get

$$\mathbb{E}_{p^n} \sup_{f \in \mathcal{F}} \mathcal{R}_p(f) - \widehat{\mathcal{R}}_p(f) \leq 2\mathrm{Rad}(\mathcal{G}, n_p). \tag{30}$$

Since $f_k(x) \geq \delta$ and $|\ell(y, f(x)) - \ell(y', f(x'))| \leq 1/\delta |f(x) - f(x')|$, by using Theorem 12 in Bartlett & Mendelson (2002), we get

$$\mathrm{Rad}(\mathcal{G}, n_p) \leq 1/\delta \cdot \mathrm{Rad}(\mathcal{F}, n_p) \tag{31}$$

Moreover, by applying the techniques in Mohri et al. (2018), we get

$$\mathrm{Rad}(\mathcal{F}, n_p) \leq \sqrt{\frac{2\mathrm{VC}(\mathcal{F}) \log n_p}{n_p}}$$

This together with Eq. (31), Eq. (30), and Eq. (29) yields

$$\mathcal{R}_p(\widehat{p}(y|x)) - \mathcal{R}_p^* \lesssim \inf_{f \in \mathcal{F}} \mathcal{R}_p(f) - \mathcal{R}_p^* + \sqrt{\frac{\mathrm{VC}(\mathcal{F}) \log n_p}{n_p}} + \sqrt{\frac{\log n_p}{n_p}} + \frac{\log n_q}{n_q}$$

with probability at least $1 - 1/n_p$. This finishes the proof. $\qquad \square$

*Proof of Theorem 5.4.* Combining Proposition B.1 and B.6, we get

$$\|\widehat{w} - w^*\|_2^2 \lesssim \inf_{f \in \mathcal{F}} \mathcal{R}_p(f) - \mathcal{R}_p^* + \sqrt{\frac{\mathrm{VC}(\mathcal{F}) \log n_p}{n_p}} + \sqrt{\frac{\log n_p}{n_p}} + \frac{\log n_p}{n_p} + 2\frac{\log n_q}{n_q}$$

$$\leq \inf_{f \in \mathcal{F}} \mathcal{R}_p(f) - \mathcal{R}_p^* + \sqrt{\frac{\mathrm{VC}(\mathcal{F}) \log n_p}{n_p}} + 2\sqrt{\frac{\log n_p}{n_p}} + 2\frac{\log n_q}{n_q}$$

with probability at least $1 - 2/n_p - 2/n_q$. This finishes the proof. $\qquad \square$

## B.4 PROOF OF THEOREM 5.5

Firstly, we present the error analysis of Theorem 5.5. To derive the excess risk of $\widetilde{q}(y|x)$, let us define

$$\widetilde{q}(k|x) = \frac{w_k^* \widehat{p}(k|x)}{\sum_{j=1}^{K} w_j^* \widehat{p}(j|x)}. \tag{32}$$

Then we are able to make the error decomposition for the excess risk of $\widehat{q}(y|x)$ in Eq. (12) as

$$\mathcal{R}_q(\widehat{q}(y|x)) - \mathcal{R}_q^* \leq \left| \mathcal{R}_q(\widehat{q}(y|x)) - \mathcal{R}_q(\widetilde{q}(y|x)) \right| + \mathcal{R}_q(\widetilde{q}(y|x)) - \mathcal{R}_q^*. \tag{33}$$

The following Propositions B.7 and B.9 provide the upper bound of these two terms in the right-hand side of Eq. (33), respectively.

**Proposition B.7.** *Let Assumptions 2.1 hold. Moreover, let $\widehat{q}(y|x)$ and $\widetilde{q}(y|x)$ be defined as in Eq. (12) and Eq. (32), respectively. Then we have*

$$\left| \mathcal{R}_q(\widehat{q}(y|x)) - \mathcal{R}_q(\widetilde{q}(y|x)) \right| \lesssim \mathcal{R}_p(\widehat{p}(k|x)) - \mathcal{R}_p^* + \|w^* - \widehat{w}\|_2^2.$$

In order to prove Proposition B.7, we first prove the following Lemma B.8.

**Lemma B.8.** *For any $a = (a_1, \ldots, a_K), x = (x_1, \ldots, x_K) \in \mathbb{R}^K$ with $a_k, x_k \in (0, 1)$, $k \in [K]$ and $\sum_{k=1}^K a_k = \sum_{k=1}^K x_k = 1$, let $f(x) := -\sum_{k=1}^K a_k \log x_k$. Then for any $x' = (x'_1, \ldots, x'_K)$ with $x'_k \in (0, 1)$, $k \in [K]$, and $\sum_{k=1}^K x'_k = 1$, we have*

$$|f(x) - f(x')| \leq \sum_{k=1}^K \left( \frac{|a_k - x_k| \cdot |x'_k - x_k|}{x_k} + \frac{a_k(x'_k - x_k)^2}{x_k(x_k \wedge x'_k)} \right).$$

*Proof of Lemma B.8.* For any $x = (x_1, \ldots, x_K), x' = (x'_1, \ldots, x'_K)$ satisfying $x_k, x'_k \in (0, 1)$, $k \in [K]$, and $\sum_{k=1}^K x_k = \sum_{k=1}^K x'_k = 1$, there holds

$$|f(x') - f(x)| = \left| \int_0^1 \left( \nabla f(x + t(x' - x)) \right)^\top (x' - x) dt \right|$$

$$= \left| \int_0^1 \nabla f(x)^\top (x' - x) dt + \int_0^1 \left( \nabla f(x + t(x' - x)) - \nabla f(x) \right)^\top (x' - x) dt \right|$$

$$\leq \left| \nabla f(x)^\top (x' - x) \right| + \left| \int_0^1 \left( \nabla f(x + t(x' - x)) - \nabla f(x) \right)^\top (x' - x) dt \right|. \quad (34)$$

For the first term in Eq. (34), there holds

$$\nabla f(x) = \nabla \left( -\sum_{k=1}^K a_k \log x_k \right) = \left( -\frac{a_1}{x_1}, \ldots, -\frac{a_K}{x_K} \right).$$

Since $\sum_{k=1}^K x_k = \sum_{k=1}^K x'_k = 1$, we then have

$$\left| \nabla f(x)^\top (x' - x) \right| = \left| -\sum_{k=1}^K \frac{a_k}{x_k}(x'_k - x_k) \right| = \left| \sum_{k=1}^K \left( 1 - \frac{a_k}{x_k} \right)(x'_k - x_k) \right|$$

$$\leq \sum_{k=1}^K \left| 1 - \frac{a_k}{x_k} \right| \cdot |x'_k - x_k| = \sum_{k=1}^K \frac{|a_k - x_k| \cdot |x'_k - x_k|}{x_k}. \quad (35)$$

For the second term in Eq. (34), we have

$$\left| \int_0^1 \left( \nabla f(x + t(x' - x)) - \nabla f(x) \right)^\top (x' - x) dt \right|$$

$$= \left| \int_0^1 \sum_{k=1}^K \left( -\frac{a_k}{x_k + t(x'_k - x_k)} + \frac{a_k}{x_k} \right)(x'_k - x_k) \, dt \right|$$

$$= \left| \int_0^1 \sum_{k=1}^K \frac{a_k t(x'_k - x_k)^2}{x_k(x_k + t(x'_k - x_k))} \, dt \right| \leq \sum_{k=1}^K \frac{a_k(x'_k - x_k)^2}{x_k(x_k \wedge x'_k)}. \quad (36)$$

Combining Eq. (34), Eq. (35), and Eq. (36), we obtain the assertion. $\qquad\square$

*Proof of Proposition B.7.* Using the law of total probability and Assumption 2.1, we obtain that for any $x \in \text{supp}(\mathrm{P}_X)$, there holds

$$\frac{q(x)}{p(x)} = \frac{\sum_{j=1}^K q(x|j)q(j)}{\sum_{j=1}^K p(x|j)p(j)} = \frac{\sum_{j=1}^K q(x|j)q(j)}{\sum_{j=1}^K q(x|j)p(j)} \leq \bigvee_{j=1}^K \frac{q(j)}{p(j)}.$$

This together with the definition of $\mathcal{R}_q$ implies

$$
\begin{aligned}
\left| \mathcal{R}_q(\widetilde{q}(y|x)) - \mathcal{R}_q(\widehat{q}(y|x)) \right| &= \left| \mathbb{E}_{x \sim q} \sum_{k=1}^{K} -q(k|x) \log \frac{\widehat{q}(k|x)}{\widetilde{q}(k|x)} \right| \\
&\leq \mathbb{E}_{x \sim q} \left| \sum_{k=1}^{K} -q(k|x) \log \frac{\widehat{q}(k|x)}{\widetilde{q}(k|x)} \right| \\
&\leq \bigvee_{k=1}^{K} \frac{q(k)}{p(k)} \cdot \mathbb{E}_{x \sim p} \left| \sum_{k=1}^{K} -q(k|x) \log \frac{\widehat{q}(k|x)}{\widetilde{q}(k|x)} \right|. \quad (37)
\end{aligned}
$$

Lemma B.8 with $a := q(y|x)$, $x' := \widehat{q}(y|x)$ and $x := \widetilde{q}(y|x)$ yields

$$
\begin{aligned}
&\mathbb{E}_{x \sim p} \left| -\sum_{k=1}^{K} q(k|x) \log \frac{\widehat{q}(k|x)}{\widetilde{q}(k|x)} \right| \\
&\leq \mathbb{E}_{x \sim p} \sum_{k=1}^{K} \frac{|q(k|x) - \widetilde{q}(k|x)| \cdot |\widetilde{q}(k|x) - \widehat{q}(k|x)|}{\widetilde{q}(k|x)} + \frac{q(k|x)|\widetilde{q}(k|x) - \widehat{q}(k|x)|^2}{\widetilde{q}(k|x)(\widetilde{q}(k|x) \wedge \widehat{q}(k|x))}.
\end{aligned}
$$

This together with Eq. (37) yields

$$
\begin{aligned}
&\left| \mathcal{R}_q(\widetilde{q}(y|x)) - \mathcal{R}_q(\widehat{q}(y|x)) \right| \\
&\leq \bigvee_{k=1}^{K} \frac{q(k)}{p(k)} \cdot \mathbb{E}_{x \sim p} \sum_{k=1}^{K} \frac{|q(k|x) - \widetilde{q}(k|x)| \cdot |\widetilde{q}(k|x) - \widehat{q}(k|x)|}{\widetilde{q}(k|x)} + \frac{q(k|x)|\widetilde{q}(k|x) - \widehat{q}(k|x)|^2}{\widetilde{q}(k|x)(\widetilde{q}(k|x) \wedge \widehat{q}(k|x))}. \quad (38)
\end{aligned}
$$

Using the definition of $\widehat{q}(y|x)$ and $\widetilde{q}(y|x)$ as in Eq. (12) and Eq. (32), respectively, we then have

$$
\begin{aligned}
\left| \widehat{q}(k|x) - \widetilde{q}(k|x) \right| &= \left| \frac{\widehat{w}_k \widehat{p}(k|x)}{\sum_{j=1}^{K} \widehat{w}_j \widehat{p}(j|x)} - \frac{w_k^* \widehat{p}(k|x)}{\sum_{j=1}^{K} w_j^* \widehat{p}(j|x)} \right| \\
&= \frac{|\sum_{j=1}^{K} \widehat{p}(j|x)(w_j^* \widehat{w}_k - w_k^* \widehat{w}_j)| \cdot \widehat{p}(k|x)}{\left( \sum_{j=1}^{K} \widehat{w}_j \widehat{p}(j|x) \right) \left( \sum_{j=1}^{K} w_j^* \widehat{p}(j|x) \right)} \\
&\leq \frac{\sum_{j=1}^{K} (|w_j^* \widehat{w}_k - w_k^* w_j^*| + |w_k^* w_j^* - w_k^* \widehat{w}_j|)}{\bigwedge_{j=1}^{k} (w_j^*)^2 / 2} \lesssim \sum_{j=1}^{K} |w_j^* - \widehat{w}_j|. \quad (39)
\end{aligned}
$$

Since for any $k \in [K]$, there exists a constant $\delta \in (0, 1/K)$ such that $p(k) \geq \delta$ and $q(k) \geq \delta$, we have $q(k) = 1 - \sum_{j \neq k} q(j) \geq 1 - (K-1)\delta$ and $p(k) = 1 - \sum_{j \neq k} p(j) \geq 1 - (K-1)\delta$. Therefore, we have

$$
\frac{\delta}{1 - (K-1)\delta} \leq w_k^* = \frac{q(k)}{p(k)} \leq \frac{1 - (K-1)\delta}{\delta}.
$$

Therefore, we can set $w_k \in [c, 1/c]$ with a constant $c \in (0, \delta/(1 - (K-1)\delta)]$. Moreover, we have

$$
\widehat{q}(k|x) = \frac{\widehat{w}_k \widehat{p}(k|x)}{\sum_{j=1}^{K} \widehat{w}_j \widehat{p}(j|x)} \geq \frac{(c^2 w_k^*) \widehat{p}(k|x)}{\sum_{j=1}^{K} (w_j^* / c^2) \widehat{p}(j|x)} = c^4 \widetilde{q}(k|x) \gtrsim \widetilde{q}(k|x) \quad (40)
$$

and

$$
\widetilde{q}(k|x) = \frac{w_k^* \widehat{p}(k|x)}{\sum_{j=1}^{K} w_j^* \widehat{p}(j|x)} \geq \frac{w_k^* \widehat{p}(k|x)}{\bigvee_{j=1}^{K} w_j^*}. \quad (41)
$$

Plugging Eq. (40) and Eq. (41) into Eq. (38), we obtain

$$
\begin{aligned}
&\left| \mathcal{R}_q(\widetilde{q}(y|x)) - \mathcal{R}_q(\widehat{q}(y|x)) \right| \\
&\leq \bigvee_{k=1}^{K} \frac{q(k)}{p(k)} \mathbb{E}_{x \sim p} \sum_{k=1}^{K} \frac{|q(k|x) - \widetilde{q}(k|x)| \cdot |\widetilde{q}(k|x) - \widehat{q}(k|x)|}{\widetilde{q}(k|x)} + \frac{q(k|x)|\widetilde{q}(k|x) - \widehat{q}(k|x)|^2}{\widetilde{q}(k|x)(\widetilde{q}(k|x) \wedge \widehat{q}(k|x))}
\end{aligned}
$$

$$\lesssim \bigvee_{k=1}^{K} \frac{q(k)}{p(k)} \mathbb{E}_{x \sim p} \sum_{k=1}^{K} \frac{|q(k|x) - \widetilde{q}(k|x)| \cdot |\widetilde{q}(k|x) - \widehat{q}(k|x)|}{w_k^* \widehat{p}(k|x) / \bigvee_{j=1}^{K} w_j^*} + \frac{q(k|x) \cdot |\widetilde{q}(k|x) - \widehat{q}(k|x)|^2}{\widetilde{q}(k|x)^2}$$

$$\lesssim \bigvee_{k=1}^{K} \frac{q(k)}{p(k)} \mathbb{E}_{x \sim p} \sum_{k=1}^{K} \frac{|q(k|x) - \widetilde{q}(k|x)| \cdot |\widetilde{q}(k|x) - \widehat{q}(k|x)|}{w_k^* \widehat{p}(k|x) / \bigvee_{j=1}^{K} w_j^*} + \frac{q(k|x) \cdot |\widetilde{q}(k|x) - \widehat{q}(k|x)|^2}{(w_k^* \widehat{p}(k|x) / \bigvee_{j=1}^{K} w_j^*)^2}$$

$$\lesssim \mathbb{E}_{x \sim p} \sum_{k=1}^{K} \frac{|q(k|x) - \widetilde{q}(k|x)| \cdot |\widetilde{q}(k|x) - \widehat{q}(k|x)|}{\widehat{p}(k|x)} + \frac{q(k|x) \cdot |\widetilde{q}(k|x) - \widehat{q}(k|x)|^2}{\widehat{p}(k|x)^2}. \tag{42}$$

Using the triangle inequality, we get

$$
\begin{aligned}
|\widetilde{q}(k|x) - q(k|x)| &= \left| \frac{w_k^* \widehat{p}(k|x)}{\sum_{j=1}^{K} w_j^* \widehat{p}(j|x)} - \frac{w_k^* p(k|x)}{\sum_{j=1}^{K} w_j^* p(j|x)} \right| \\
&= \frac{w_k^* \cdot |\sum_{j=1}^{K} w_j^* (p(j|x)\widehat{p}(k|x) - p(k|x)\widehat{p}(j|x))|}{\left( \sum_{j=1}^{K} w_j^* \widehat{p}(j|x) \right) \left( \sum_{j=1}^{K} w_j^* p(j|x) \right)} \\
&\leq \frac{w_k^* \sum_{j=1}^{K} w_j^* (p(j|x)|\widehat{p}(k|x) - p(k|x)| + p(k|x)|p(j|x) - \widehat{p}(j|x)|)}{(\bigwedge_{j=1}^{K} w_j^*)^2} \\
&\lesssim \sum_{j=1}^{K} |\widehat{p}(j|x) - p(j|x)|. \tag{43}
\end{aligned}
$$

Moreover, we have

$$q(k|x) = \frac{w_k^* p(k|x)}{\sum_{j=1}^{K} w_j^* p(j|x)} \leq \frac{w_k^* p(k|x)}{\bigwedge_{j=1}^{K} w_j^*} \lesssim p(k|x). \tag{44}$$

Combining Eq. (42), Eq. (39), Eq. (43), and Eq. (44), and using the Cauchy-Schwarz inequality, we obtain

$$
\begin{aligned}
&\left| \mathcal{R}_q(\widetilde{q}(y|x)) - \mathcal{R}_q(\widehat{q}(y|x)) \right| \\
&\lesssim \mathbb{E}_{x \sim p} \sum_{k=1}^{K} \frac{(\sum_j |\widehat{p}(j|x) - p(j|x)|) \cdot (\widehat{p}(k|x) \sum_j |w_j^* - \widehat{w}_j|)}{\widehat{p}(k|x)} + \frac{p(k|x)(\widehat{p}(k|x) \sum_j |w_j^* - \widehat{w}_j|)^2}{\widehat{p}(k|x)^2} \\
&\lesssim \mathbb{E}_{x \sim p} \left( \left( \sum_{j=1}^{K} |\widehat{p}(j|x) - p(j|x)| \right) \cdot \left( \sum_{j=1}^{K} |w_j^* - \widehat{w}_j| \right) + \left( \sum_{j=1}^{K} |w_j^* - \widehat{w}_j| \right)^2 \right) \\
&\leq \sqrt{\mathbb{E}_{x \sim p} \left( \sum_{j=1}^{K} |\widehat{p}(j|x) - p(j|x)| \right)^2 \cdot \left( \sum_{j=1}^{K} |w_j^* - \widehat{w}_j| \right)^2} + \left( \sum_{j=1}^{K} |w_j^* - \widehat{w}_j| \right)^2 \\
&\lesssim \mathbb{E}_{x \sim p} \left( \sum_{j=1}^{K} |\widehat{p}(j|x) - p(j|x)| \right)^2 + \left( \sum_{j=1}^{K} |w_j^* - \widehat{w}_j| \right)^2 \\
&\lesssim \sum_{j=1}^{K} \left( \mathbb{E}_{x \sim p} |\widehat{p}(j|x) - p(j|x)|^2 + |w_j^* - \widehat{w}_j|^2 \right) \\
&= \mathcal{R}_p(\widehat{p}(k|x)) - \mathcal{R}_p^* + \|w^* - \widehat{w}\|_2^2,
\end{aligned}
$$

where the last inequality follows from Lemma B.4. This finishes the proof. $\qquad \square$

**Proposition B.9.** *Let Assumptions 2.1 hold. Moreover, let $\widetilde{q}(y|x)$ and $\widehat{p}(y|x)$ be as in Eq. (32) and Eq. (17), respectively. Then we have*

$$\mathcal{R}_q(\widetilde{q}(y|x)) - \mathcal{R}_q^* \lesssim \mathcal{R}_p(\widehat{p}(y|x)) - \mathcal{R}_p^*.$$

To prove Proposition B.9, we need the following basic lemma.

**Lemma B.10.** *For any* $k \in [K]$*, let* $a_k, b_k > 0$*,* $c_k, x_k \in (0,1)$ *satisfy* $\sum_{k=1}^{K} c_k = 1$ *and* $\sum_{k=1}^{K} x_k = 1$*. Then we have*

$$\sum_{k=1}^{K} \frac{a_k c_k}{\bigvee_{j=1}^{K} a_j} \log \frac{c_k / (\sum_{j=1}^{K} a_j c_j)}{x_k / (\sum_{j=1}^{K} a_j x_j)} \leq \sum_{k=1}^{K} \left( c_k \log \left( \frac{c_k}{x_k} \right) \right).$$

*Proof of Lemma B.10.* Let the function $h : (0,1)^K \to \mathbb{R}$ be defined by

$$h(x) := h(x_1, \ldots, x_K)$$
$$:= \sum_{k=1}^{K} \left( \frac{a_k c_k}{\bigvee_{j=1}^{K} a_j} \log \frac{x_k / (\sum_{j=1}^{K} a_j x_j)}{c_k / (\sum_{j=1}^{K} a_j c_j)} + c_k \log \left( \frac{c_k}{x_k} \right) \right) + \lambda \left( \sum_{k=1}^{K} x_k - 1 \right),$$

where $\lambda > 0$ is the Lagrange multiplier. Then it suffices to prove that $h(x) \geq h(c) = 0$ for any $x$ satisfying $\sum_{j=1}^{K} x_j = 1$ and $0 < x_j < 1$, $j \in [K]$. Taking the partial derivative of $h(x)$ w.r.t. $x_k$ and setting it to be zero, i.e.

$$\frac{\partial h(x)}{\partial x_k} = \frac{a_k c_k}{\bigvee_{j=1}^{K} a_j} \cdot \frac{1}{x_k} - \sum_{\ell=1}^{K} \frac{a_\ell c_\ell}{\bigvee_{j=1}^{K} a_j} \cdot \frac{a_k}{\sum_{j=1}^{K} a_j x_j} - \frac{c_k}{x_k} + \lambda = 0, \qquad k = 1, \ldots, K, \quad (45)$$

which is equivalent to

$$\left( \frac{a_k}{\bigvee_{j=1}^{K} a_j} - 1 \right) c_k - \frac{a_k x_k}{\bigvee_{j=1}^{K} a_j} \cdot \frac{\sum_{j=1}^{K} a_j c_j}{\sum_{j=1}^{K} a_j x_j} + \lambda x_k = 0, \qquad k = 1, \ldots, K.$$

Taking the summation from $k = 1$ to $K$ and using $\sum_{k=1}^{K} x_k = \sum_{k=1}^{K} c_k = 1$, we get

$$\sum_{k=1}^{K} \frac{a_k c_k}{\bigvee_{j=1}^{K} a_j} - 1 - \frac{\sum_{j=1}^{K} a_j c_j}{\bigvee_{j=1}^{K} a_j} + \lambda = 0,$$

which implies $\lambda = 1$. This together with Eq. (45) yields

$$\left( 1 - \frac{a_k}{\bigvee_{j=1}^{K} a_j} \right) \left( 1 - \frac{c_k}{x_k} \right) + \frac{a_k}{\bigvee_{j=1}^{K} a_j} \left( 1 - \frac{\sum_{j=1}^{K} a_j c_j}{\sum_{j=1}^{K} a_j x_j} \right) = 0, \qquad k = 1, \ldots, K. \quad (46)$$

If $\sum_{j=1}^{K} a_j c_j \neq \sum_{j=1}^{K} a_j x_j$, then there must exist some $\ell \in [K]$ such that $c_\ell \neq x_\ell$. Since $\sum_{j=1}^{K} x_j = \sum_{j=1}^{K} c_j = 1$, there exist some $i, j \in [K]$ such that $c_i > x_i$ and $c_j < x_j$. Without loss of generality, we assume that $\sum_{j=1}^{K} a_j c_j > \sum_{j=1}^{K} a_j x_j$. Therefore, we have both $1 - c_i / x_i < 0$ and $1 - \sum_{j=1}^{K} a_j c_j / \sum_{j=1}^{K} a_j x_j < 0$. Thus we have

$$\left( 1 - \frac{a_i}{\bigvee_{j=1}^{K} a_j} \right) \left( 1 - \frac{c_i}{x_i} \right) + \frac{a_i}{\bigvee_{j=1}^{K} a_j} \left( 1 - \frac{\sum_{j=1}^{K} a_j c_j}{\sum_{j=1}^{K} a_j x_j} \right) < 0$$

which contradicts with Eq. (46) for $k = i$. Therefore, we have $\sum_{j=1}^{K} a_j c_j = \sum_{j=1}^{K} a_j x_j$, which together with Eq. (46) implies $c_k = a_k$ for any $k \in [K]$. Thus $h(x)$ attains its minimum at the point $x = (c_1, \ldots, c_k)$ under the constraint $\sum_{k=1}^{K} x_k = 1$, i.e., $h(x) \geq h(c) = 0$ holds for any $x$ with $\sum_{j=1}^{K} x_j = 1$ and $x_j \in (0,1)$ for any $j \in [K]$, which finishes the proof. $\square$

*Proof of Proposition B.9.* Using the law of total probability and Assumption 2.1, we obtain that for any $x \in \mathrm{supp}(\mathrm{P}_X)$, there holds

$$\frac{q(x)}{p(x)} = \frac{\sum_{j=1}^{K} q(x|j) q(j)}{\sum_{j=1}^{K} p(x|j) p(j)} = \frac{\sum_{j=1}^{K} q(x|j) q(j)}{\sum_{j=1}^{K} q(x|j) p(j)} \leq \bigvee_{j=1}^{K} \frac{q(j)}{p(j)}. \quad (47)$$

Therefore, we have

$$\mathcal{R}_q(\widetilde{q}(y|x)) - \mathcal{R}_q^* = \mathbb{E}_{X\sim q} \sum_{k=1}^{K} q(k|X) \log \frac{q(k|X)}{\widetilde{q}(k|X)}$$

$$\leq \bigvee_{k=1}^{K} \frac{q(k)}{p(k)} \cdot \mathbb{E}_{X\sim p} \sum_{k=1}^{K} q(k|X) \log \frac{q(k|X)}{\widetilde{q}(k|X)}. \tag{48}$$

Using the definitions of $\widetilde{q}(y|x)$ in Eq. (32), respectively, and Lemma B.10 with $a_k := q(k)/p(k)$, $b_k := (1 - q(k))/(1 - p(k))$, $c_k := p(k|x)$, and $x_k := \widetilde{p}(k|x)$, we obtain

$$\sum_{k=1}^{K} q(k|X) \log \frac{q(k|X)}{\widetilde{q}(k|X)}$$

$$= \sum_{k=1}^{K} \frac{(q(k)/p(k))p(k|x)}{\sum_{j=1}^{K}(q(j)/p(j))p(j|x)} \cdot \log \frac{(q(k)/p(k))p(k|x)/(\sum_{j=1}^{K}(q(j)/p(j))p(j|x))}{((q(k)/p(k))\widehat{p}(k|x)/\sum_{j=1}^{K}(q(j)/p(j))\widehat{p}(j|x))}$$

$$\leq \frac{\bigvee_{k=1}^{K} q(k)/p(k)}{\bigwedge_{k=1}^{K} q(k)/p(k)} \cdot \sum_{k=1}^{K} \frac{(q(k)/p(k))p(k|x)}{\bigvee_{k=1}^{K} q(k)/p(k)} \cdot \log \frac{(q(k)/p(k))p(k|x)/(\sum_{j=1}^{K}(q(j)/p(j))p(j|x))}{((q(k)/p(k))\widehat{p}(k|x)/\sum_{j=1}^{K}(q(j)/p(j))\widehat{p}(j|x))}$$

$$\leq \frac{\bigvee_{k=1}^{K} q(k)/p(k)}{\bigwedge_{k=1}^{K} q(k)/p(k)} \cdot \sum_{k=1}^{K} p(k|x) \log \frac{p(k|x)}{\widehat{p}(k|x)}.$$

This together with Eq. (48) yields

$$\mathcal{R}_q(\widetilde{q}(y|x)) - \mathcal{R}_q^* \leq \frac{\left(\bigvee_{k=1}^{K} q(k)/p(k)\right)^2}{\bigwedge_{k=1}^{K} q(k)/p(k)} \left(\mathcal{R}_p(\widehat{p}(y|x)) - \mathcal{R}_p^*\right) \lesssim \mathcal{R}_p(\widehat{p}(y|x)) - \mathcal{R}_p^*,$$

which proves the assertion. □

With these preparation, we give the proof of Theorem 5.5.

*Proof of Theorem 5.5.* Combining Eq. (33), Propositions B.7, B.9 and B.6, we obtain

$$\mathcal{R}_q(\widehat{q}(y|x)) - \mathcal{R}_q^* \leq 2(\mathcal{R}_p(\widehat{p}(k|x)) - \mathcal{R}_p^*) + \|w^* - \widehat{w}\|_2^2$$

$$\lesssim 3(\mathcal{R}_p(\widehat{p}(k|x)) - \mathcal{R}_p^*) + \frac{\log n_q}{n_q} + \frac{\log n_p}{n_p}$$

$$\leq 3\left( \inf_{f\in\mathcal{F}} \mathcal{R}_p(f) - \mathcal{R}_p^* + \sqrt{\frac{\mathrm{VC}(\mathcal{F})\log n_p}{n_p}} + \sqrt{\frac{\log n_p}{n_p} + \frac{\log n_q}{n_q}} \right) + \frac{\log n_q}{n_q} + \frac{\log n_p}{n_p}$$

$$\leq 3\left( \inf_{f\in\mathcal{F}} \mathcal{R}_p(f) - \mathcal{R}_p^* + \sqrt{\frac{\mathrm{VC}(\mathcal{F})\log n_p}{n_p}} + 2\sqrt{\frac{\log n_p}{n_p}} + \frac{2\log n_q}{n_q} \right)$$

with probability at least $1 - 2/n_p - 2/n_q$. This finishes the proof. □

## C    Supplementaries for Experiments

### C.1    Evaluation Metrics

We consider the following three metrics for the evaluations of label shift adaptation problems. To be specific, the first one `ACC` is the classification accuracy in the target domain. The other two metrics, `MSE_PROP` and `MSE_EVEN`, are used to measure the estimation error of $\widehat{w}$ for estimating the class probability ratio $w^*$.

**(i)** `ACC` is the accuracy of domain-adapted model predictions in the target domain. Mathematically speaking, `ACC` equals $\frac{1}{n_q} \sum_{i=n_p+1}^{n_p+n_q} \mathbf{1}\{\widehat{h}_q(X_i) = Y_i\}$, where $\{Y_i\}_{n_p+1}^{n_p+n_q}$ are the true labels of $D_q^u$.

**(ii)** `MSE_PROP` is the mean squared error weighted by the test set proportion $q(k)$ and calculated by taking the weighted average of the squared differences between the estimated weight $\widehat{w}_k$ and true weights $w_k^*$, i.e. $\sum_k q(k)(\widehat{w}_k - w_k^*)^2$.

**(iii)** `MSE_EVEN` is the mean squared error with equal weight and calculated by taking the uniform average of the squared differences between the estimated and true weights, i.e., $\frac{1}{K}\sum_k(\widehat{w}_k - w_k)^2$, where the number of classes is denoted by $K$.

Note that larger `ACC` and smaller `MSE_PROP` and `MSE_EVEN` indicate better performance.

### C.2 IMPLEMENTATION DETAILS OF ALL ALGORITHMS

The implementation of the CPMCN algorithm is divided into three parts:

- Training the network on the first 50,000 training points of the source domain data. The training code and parameters can be found in the `obtaining_predictions` folder at `https://github.com/kundajelab/labelshiftexperiments/tree/master/notebooks/obtaining_predictions`. The MNIST, CIFAR10, and CIFAR100 folders correspond to the networks for the respective datasets. We use the codes and hyperparameter values from the above github repository to train the network.

- We utilize to correct the output of the network, resulting in the calibrated neural network. The calibrated method Bias-Corrected Temperature Scaling (BCTS) is implemented based on the code in `https://github.com/kundajelab/abstention/blob/master/abstention/calibration.py`. It is invoked using the `TempScaling(bias_positions='all')` method. The last 10,000 data points in the training set are used as a validation set to train the calibration parameters of BCTS.

- Based on the calibrated neural network, we follow Algorithm 1 to obtain the target domain's classifier. For some target samples $X_i$, the denominator $\sum_{k=1}^{K} w_k\widehat{p}(k|X_i)$ in Eq. (10) may be close to zero, leading to the computational issue in calculating $\widehat{p}_q^w(y)$. Conversely, we observe that $w_y\widehat{p}_q^w(y) = n_q^{-1}\sum_{i=n_p+1}^{n_p+n_q} w_y\widehat{p}(y|X_i)/(\sum_{k\in[K]} w_k\widehat{p}(k|X_i))$ falls within the range of 0 to 1. Fortunately, it is easy to see that $w^*$ is also the solution of $w_y p(y) = w_y p_q^w(y)$. Therefore, to address the computational issue, we introduce $\sum_{y\in[K]} |w_y\widehat{p}(y) - w_y\widehat{p}_q^w(y)|^2$ as the objective function with respect to $w$. We optimize the function by using a variant of sequential minimal optimization (Platt, 1998) combined with Nelder-Mead method (Nelder & Mead, 1965).

To implement the compared methods BBSL, RLLS and EM, we use the code in `https://github.com/kundajelab/labelshiftexperiments` provided by Alexandari et al. (2020). For the compared method ELSA, we implement it in Python and use L-BFGS-B (Zhu et al., 1997) as the optimization method with the solution of BBSL as its initial value.

### C.3 ADDITIONAL RESULTS ON THE CIFAR100, MNIST AND CIFAR10 DATASETS

| $\rho$ | 0.01 | | | 0.02 | | | 0.05 | | |
|---|---|---|---|---|---|---|---|---|---|
| Metrics | MSE_EVEN | MSE_PROP | ACC (%) | MSE_EVEN | MSE_PROP | ACC (%) | MSE_EVEN | MSE_PROP | ACC (%) |
| LTF | 8.41e+0 (9.12e+0) | 8.40e+1 (8.51e+1) | 84.45 (3.38) | 4.93e+0 (6.23e+0) | 5.21e+1 (6.81e+1) | 85.20 (3.04) | 5.11e+0 (3.43e+0) | 5.26e+1 (3.64e+1) | 85.25 (1.06) |
| BBSL | 5.61e-1 (1.95e+2) | 3.93e+0 (9.12e+0) | 89.15 (0.33) | 5.56e-1 (2.07e+2) | 3.79e+0 (9.04e+0) | 88.75 (0.33) | 5.56e-1 (2.18e+2) | 3.65e+0 (8.77e+0) | 88.25 (0.32) |
| RLLS | 4.26e-1 (4.75e-2) | 3.56e+0 (4.91e-1) | 89.45 (0.14) | 4.17e-1 (4.60e-2) | 3.30e+0 (4.74e-1) | 89.15 (0.14) | 4.21e-1 (4.27e-2) | 3.15e+0 (4.24e-1) | 88.35 (0.14) |
| ELSA | 1.90e+0 (7.51e+0) | 6.85e+0 (1.50e+0) | 89.85 (0.28) | 1.82e+0 (6.32e+0) | 7.11e+0 (1.42e+0) | 89.40 (0.27) | 1.82e+0 (5.52e+0) | 6.74e+0 (1.32e+0) | 88.65 (0.27) |
| EM | 1.17e-1 (9.63e-3) | 9.32e-1 (9.08e-2) | 90.60 (0.12) | 1.23e-1 (9.39e-3) | 9.30e-1 (8.81e-2) | 90.23 (0.12) | 1.26e-1 (8.74e-3) | 9.76e-1 (7.92e-2) | 89.65 (0.12) |
| CPMCN | **1.06e-1** **(9.77e-3)** | **8.06e-1** **(9.19e-2)** | **90.85** **(0.12)** | **1.07e-1** **(9.35e-3)** | **8.69e-1** **(8.76e-2)** | **90.50** **(0.11)** | **1.09e-1** **(8.53e-3)** | **8.65e-1** **(7.57e-2)** | **90.00** **(0.12)** |
| $\rho$ | 0.6 | | | 0.8 | | | 0.9 | | |
| Metrics | MSE_EVEN | MSE_PROP | ACC (%) | MSE_EVEN | MSE_PROP | ACC (%) | MSE_EVEN | MSE_PROP | ACC (%) |
| LTF | 4.41e+1 (8.97e+0) | 1.44e+3 (2.06e+2) | 72.32 (2.21) | 5.07e+1 (6.27e+0) | 3.53e+3 (3.38e+2) | 69.50 (0.80) | 5.78e+1 (4.77e+0) | 4.86e+3 (3.80e+2) | 66.30 (1.57) |
| BBSL | 2.26e+0 (4.31e+1) | 3.56e+1 (1.56e+1) | 85.15 (0.35) | 3.62e+0 (2.86e+1) | 8.48e+1 (3.62e+1) | 85.20 (0.46) | 4.44e+0 (1.56e+1) | 1.25e+2 (5.21e+1) | 85.93 (0.49) |
| RLLS | 5.09e+0 (1.40e+0) | 2.44e+2 (8.32e+1) | 83.40 (0.49) | 9.71e+0 (2.68e+0) | 6.45e+2 (2.14e+2) | 83.05 (0.67) | 1.22e+1 (3.42e+0) | 9.32e+2 (3.06e+2) | 83.40 (0.75) |
| ELSA | 6.23e+0 (9.05e+0) | 5.58e+1 (3.69e+1) | 85.18 (0.65) | 1.01e+1 (1.40e+1) | 1.19e+2 (8.89e+1) | 84.68 (0.93) | 1.25e+1 (1.72e+1) | 1.80e+2 (1.28e+2) | 85.20 (0.99) |
| EM | **6.86e-1** **(7.70e-2)** | 2.99e+1 (3.37e+0) | 87.75 (0.25) | **1.07e+0** **(1.20e-1)** | 5.53e+1 (6.96e+0) | 89.90 (0.29) | 1.27e+0 (1.38e-1) | 7.69e+1 (9.09e+0) | 91.20 (0.32) |
| CPMCN | 6.95e-1 (7.83e-2) | **2.82e+1** **(3.45e+0)** | **87.98** **(0.25)** | **1.07e+0** **(1.22e-1)** | 5.70e+1 (6.97e+0) | **90.30** **(0.31)** | **1.04e+0** **(1.50e-1)** | **6.02e+1** **(9.63e+1)** | **92.18** **(0.32)** |

Table 2: Performance on CIFAR100 under the Tweak-one shift. For each method, the first line is the median of the corresponding metrics among 100 repeating experiments and the second line is the standard deviation of these 100 metrics. The best results are marked in **bold**.

| $\alpha$ | 0.1 | | | 0.2 | | | 0.5 | | |
|---|---|---|---|---|---|---|---|---|---|
| Metrics | MSE_EVEN | MSE_PROP | ACC (%) | MSE_EVEN | MSE_PROP | ACC (%) | MSE_EVEN | MSE_PROP | ACC (%) |
| LTF | 3.27e-2 (1.25e-2) | 9.47e-2 (8.20e-2) | 97.70 (0.46) | 3.72e-2 (7.29e-3) | 1.43e-1 (3.97e-2) | 96.30 (0.39) | 1.94e-2 (4.98e-3) | 3.38e-2 (1.33e-2) | 93.63 (0.39) |
| BBSL | 4.13e-3 (6.34e-4) | 8.96e-3 (3.99e-3) | 97.50 (0.15) | 3.54e-3 (5.34e-4) | 7.60e-3 (3.10e-3) | 95.75 (0.16) | 2.83e-3 (2.20e-4) | 4.43e-3 (6.52e-4) | **94.03** **(0.15)** |
| RLLS | 3.99e-3 (6.35e-4) | 9.33e-3 (4.01e-3) | 97.50 (0.15) | 3.61e-3 (5.36e-4) | 7.92e-3 (3.11e-3) | 95.75 (0.16) | 2.83e-3 (2.19e-4) | 4.43e-3 (6.47e-4) | 93.97 (0.15) |
| ELSA | 8.81e-2 (4.40e+2) | 1.04e-1 (4.08e+3) | 96.05 (2.29) | 2.01e-1 (5.77e+3) | 1.43e-1 (6.45e+3) | 93.85 (1.32) | 2.22e-1 (1.98e+1) | 1.86e-1 (5.23e+0) | 91.78 (0.74) |
| EM | 8.49e-4 (1.34e-4) | **2.14e-3** **(6.44e-4)** | 97.55 (0.15) | **1.03e-3** **(2.41e-4)** | **2.09e-3** **(9.74e-4)** | 95.88 (0.17) | 1.25e-3 **(1.65e-4)** | 2.03e-3 **(5.44e-4)** | 93.98 (0.16) |
| CPMCN | **8.42e-4** **(1.34e-4)** | 2.14e-3 **(6.38e-4)** | 97.55 (0.15) | **1.03e-3** **(2.41e-4)** | 2.11e-3 (9.74e-4) | 95.88 (0.17) | 1.26e-3 (1.64e-4) | 2.04e-3 (5.43e-4) | 93.98 (0.16) |
| $\alpha$ | 0.8 | | | 1.0 | | | 10 | | |
| Metrics | MSE_EVEN | MSE_PROP | ACC (%) | MSE_EVEN | MSE_PROP | ACC (%) | MSE_EVEN | MSE_PROP | ACC (%) |
| LTF | 1.93e-2 (4.29e-3) | 2.96e-2 (1.11e-2) | 93.47 (0.34) | 2.10e-2 (6.38e-3) | 2.91e-2 (2.05e-2) | 92.73 (0.27) | 9.94e-3 (1.37e-3) | 9.62e-3 (1.20e-3) | 91.70 (0.08) |
| BBSL | 1.90e-3 (2.02e-4) | 2.75e-3 (6.07e-4) | 93.20 (0.13) | 2.28e-3 (1.65e-4) | 3.03e-3 (4.71e-4) | **93.20** **(0.12)** | 1.68e-3 (1.03e-4) | **1.69e-3** **(1.23e-4)** | 91.70 (0.07) |
| RLLS | 1.90e-3 (2.02e-4) | 2.76e-3 (6.07e-4) | 93.20 (0.13) | 2.26e-3 (1.65e-4) | 3.0e-3 (4.69e-4) | **93.20** **(0.12)** | 1.68e-3 (1.03e-4) | **1.69e-3** **(1.23e-4)** | 91.70 (0.07) |
| ELSA | 6.76e-2 (4.02e+1) | 4.90e-2 (1.18e+1) | 92.23 (0.60) | 1.98e-1 (3.84e+0) | 1.63e-1 (5.94e-1) | 91.88 (0.34) | 2.00e-2 (6.91e-3) | 1.88e-2 (7.67e-3) | 91.53 (0.07) |
| EM | **1.46e-3** **(1.25e-4)** | 2.10e-3 (2.68e-4) | **93.25** **(0.13)** | 1.65e-3 (1.18e-4) | **2.49e-3** **(2.62e-4)** | 93.20 (0.12) | 1.57e-3 (1.12e-4) | 1.71e-3 (1.33e-4) | **91.73** **(0.07)** |
| CPMCN | **1.46e-3** **(1.25e-4)** | **2.07e-3** **(2.69e-4)** | **93.25** **(0.13)** | **1.64e-3** **(1.18e-4)** | 2.51e-3 (2.62e-4) | 93.20 (0.12) | **1.56e-3** **(1.12e-4)** | 1.71e-3 (1.32e-4) | **91.73** **(0.07)** |

Table 3: Performance on MNIST under the Dirichlet shift. For each method, the first line is the median of the corresponding metrics among 100 repeating experiments and the second line is the standard derivation of these 100 metrics. The best results are marked as **bold**.

| $\rho$ | 0.01 | | | 0.02 | | | 0.05 | | |
|---|---|---|---|---|---|---|---|---|---|
| Metrics | MSE_EVEN | MSE_PROP | ACC (%) | MSE_EVEN | MSE_PROP | ACC (%) | MSE_EVEN | MSE_PROP | ACC (%) |
| LTF | 8.08e-2 | 7.68e-2 | 91.55 | 5.76e-2 | 4.81e-2 | 91.90 | 3.32e-2 | 3.43e-2 | 91.70 |
| | (8.83e-3) | (9.68e-3) | (0.16) | (4.97e-3) | (5.14e-3) | (0.16) | (4.88e-3) | (5.18e-3) | (0.12) |
| BBSL | 1.58e-3 | 1.66e-3 | **92.05** | 1.54e-3 | 1.58e-3 | **91.98** | 1.61e-3 | 1.65e-3 | 91.73 |
| | (9.36e-5) | (1.01e-4) | **(0.06)** | (8.96e-5) | (9.64e-5) | **(0.06)** | (8.38e-5) | (8.76e-5) | (0.06) |
| RLLS | 1.58e-3 | 1.66e-3 | **92.05** | 1.54e-3 | 1.58e-3 | **91.98** | 1.61e-3 | 1.65e-3 | 91.73 |
| | (9.36e-5) | (1.01e-4) | **(0.06)** | (8.96e-5) | (9.64e-5) | **(0.06)** | (8.38e-5) | (8.76e-5) | (0.06) |
| ELSA | 2.45e-3 | 2.69e-3 | 92.00 | 2.29e-3 | 2.48e-3 | 91.95 | 2.59e-3 | 2.69e-3 | **91.75** |
| | (8.93e-3) | (2.21e-3) | (0.07) | (4.66e-4) | (5.08e-4) | (0.06) | (4.40e-4) | (4.64e-4) | **(0.06)** |
| EM | **1.53e-3** | **1.64e-3** | 92.00 | **1.50e-3** | **1.56e-3** | 91.93 | 1.54e-3 | **1.60e-3** | 91.73 |
| | **(9.58e-5)** | **(1.05e-4)** | (0.06) | **(9.18e-5)** | **(1.0e-4)** | (0.06) | (9.09e-5) | **(9.53e-5)** | (0.06) |
| CPMCN | **1.53e-3** | **1.64e-3** | 92.00 | 1.51e-3 | **1.56e-3** | 91.93 | **1.53e-3** | **1.60e-3** | 91.73 |
| | **(9.56e-5)** | **(1.05e-4)** | (0.06) | (9.15e-5) | **(9.98e-5)** | (0.06) | **(9.06e-5)** | **(9.51e-5)** | (0.06) |
| $\rho$ | 0.6 | | | 0.8 | | | 0.9 | | |
| Metrics | MSE_EVEN | MSE_PROP | ACC (%) | MSE_EVEN | MSE_PROP | ACC (%) | MSE_EVEN | MSE_PROP | ACC (%) |
| LTF | 2.34e-2 | 7.97e-2 | 93.53 | 3.09e-2 | 1.97e-1 | 95.50 | 3.44e-2 | 1.84e-1 | 96.85 |
| | (3.70e-3) | (1.43e-2) | (0.15) | (4.09e-3) | (2.22e-2) | (0.10) | (5.28e-3) | (3.61e-2) | (0.08) |
| BBSL | 3.89e-3 | 7.45e-3 | 93.58 | 6.45e-3 | 1.29e-2 | 95.35 | 6.23e-3 | 1.81e-2 | 96.38 |
| | (5.56e-4) | (2.49e-3) | (0.06) | (8.25e-4) | (5.15e-3) | (0.07) | (8.69e-4) | (7.40e-3) | (0.06) |
| RLLS | 3.89e-3 | 7.45e-3 | 93.58 | 6.45e-3 | 1.29e-2 | 95.35 | 6.03e-3 | 1.78e-2 | 96.35 |
| | (5.56e-4) | (2.49e-3) | (0.06) | (8.14e-4) | (5.05e-3) | (0.07) | (7.82e-4) | (6.57e-3) | (0.07) |
| ELSA | 1.98e-3 | 2.65e-3 | 93.58 | 1.87e-3 | 3.74e-3 | 95.55 | 7.27e-3 | 7.00e-3 | 96.43 |
| | (3.26e-4) | (1.53e-3) | (0.06) | (1.66e-3) | (8.13e-3) | (0.08) | (1.70e+2) | (1.13e+2) | (0.13) |
| EM | 1.11e-3 | **1.72e-3** | **93.65** | 8.64e-4 | **1.52e-3** | 95.60 | 5.97e-4 | **1.39e-3** | 97.00 |
| | (1.44e-4) | **(7.49e-4)** | **(0.06)** | (8.46e-5) | **(5.68e-4)** | **(0.05)** | **(4.77e-5)** | **(3.54e-4)** | **(0.04)** |
| CPMCN | **1.10e-3** | 1.75e-3 | **93.65** | **8.58e-4** | 1.57e-3 | **95.60** | **5.97e-4** | 1.43e-3 | **97.00** |
| | **(1.44e-4)** | (7.51e-4) | **(0.06)** | **(8.45e-5)** | (5.67e-4) | **(0.05)** | **(4.78e-5)** | (3.55e-4) | **(0.04)** |

Table 4: Performance on MNIST under the tweak-one shift. For each method, the first line is the median of the corresponding metrics among 100 repeating experiments and the second line is the standard derivation of these 100 metrics. The best results are marked as **bold**.

| $\alpha$ | 0.1 | | | 0.2 | | | 0.5 | | |
|---|---|---|---|---|---|---|---|---|---|
| Metrics | MSE_EVEN | MSE_PROP | ACC (%) | MSE_EVEN | MSE_PROP | ACC (%) | MSE_EVEN | MSE_PROP | ACC (%) |
| LTF | 6.28e-3 | 9.24e-3 | 97.65 | 1.26e-2 | 1.58e-2 | 94.75 | 3.38e-2 | 4.06e-2 | 92.55 |
| | (2.66e-3) | (4.91e-3) | (0.57) | (1.90e-2) | (6.05e-2) | (1.49) | (9.48e-3) | (2.49e-2) | (1.04) |
| BBSL | 2.88e-3 | 8.60e-3 | 97.60 | 2.80e-3 | 6.57e-3 | 96.12 | 2.40e-3 | 3.68e-3 | 93.75 |
| | (1.82e-3) | (1.54e-2) | (0.20) | (5.47e-4) | (2.94e-3) | (0.20) | (8.78e-4) | (4.44e-3) | (0.17) |
| RLLS | 2.84e-3 | 8.23e-3 | 97.58 | 2.82e-3 | 6.53e-3 | 96.17 | 2.44e-3 | 3.66e-3 | 93.75 |
| | (1.53e-3) | (1.25e-2) | (0.20) | (5.36e-4) | (2.90e-3) | (0.20) | (8.76e-4) | (4.44e-3) | (0.17) |
| ELSA | 4.02e-1 | 5.85e-1 | 95.60 | 1.58e-1 | 1.34e-1 | 94.52 | 1.87e-1 | 1.52e-1 | 93.08 |
| | (2.74e+2) | (2.08e+2) | (1.68) | (9.67e+1) | (2.17e+1) | (1.09) | (1.26e-1) | (3.29e-1) | (0.42) |
| EM | **3.68e-4** | 1.36e-3 | **97.70** | 9.29e-4 | **2.15e-3** | **96.18** | 1.37e-3 | 2.11e-3 | **93.80** |
| | **(3.35e-4)** | (1.73e-3) | **(0.20)** | (3.02e-4) | **(1.30e-3)** | **(0.20)** | (1.77e-4) | (5.71e-4) | **(0.17)** |
| CPMCN | 3.69e-4 | **1.33e-3** | **97.70** | **9.28e-4** | 2.16e-3 | **96.18** | **1.37e-3** | **2.10e-3** | **93.80** |
| | (3.32e-4) | **(1.71e-3)** | **(0.20)** | **(3.01e-4)** | (1.30e-3) | **(0.20)** | **(1.77e-4)** | **(5.69e-4)** | **(0.17)** |
| $\alpha$ | 0.8 | | | 1.0 | | | 10 | | |
| Metrics | MSE_EVEN | MSE_PROP | ACC (%) | MSE_EVEN | MSE_PROP | ACC (%) | MSE_EVEN | MSE_PROP | ACC (%) |
| LTF | 3.59e-2 | 3.09e-2 | 92.85 | 3.40e-2 | 6.23e-2 | 91.65 | 1.58e-2 | 1.64e-2 | 90.73 |
| | (8.64e-3) | (1.69e-2) | (1.15) | (9.83e-4) | (1.36e-2) | (0.75) | (4.80e-4) | (1.56e-3) | (0.02) |
| BBSL | 2.35e-3 | 2.93e-3 | 92.95 | 1.71e-3 | 2.58e-3 | 92.90 | 1.76e-3 | 1.76e-3 | 90.80 |
| | (5.75e-4) | (2.76e-3) | (0.16) | (3.09e-4) | (9.77e-4) | (0.16) | (1.51e-4) | (1.74e-4) | (0.09) |
| RLLS | 2.35e-3 | 2.94e-3 | 93.00 | 1.69e-3 | 2.50e-3 | 92.90 | 1.76e-3 | 1.76e-3 | 90.80 |
| | (5.66e-4) | (2.74e-3) | (0.16) | (3.08e-4) | (9.69e-4) | (0.16) | (1.51e-4) | (1.74e-4) | (0.09) |
| ELSA | 6.87e-2 | 5.78e-2 | 92.23 | 1.71e-1 | 1.58e-1 | 92.23 | 2.37e-2 | 2.17e-2 | 90.75 |
| | (8.30e-2) | (1.90e-1) | (0.45) | (8.17e-2) | (1.81e-1) | (0.29) | (7.18e-3) | (8.04e-3) | (0.09) |
| EM | **1.43e-3** | **2.01e-3** | **93.18** | 1.26e-3 | 1.52e-3 | **92.98** | **1.61e-3** | **1.68e-3** | **90.85** |
| | **(2.32e-4)** | **(8.18e-4)** | **(0.16)** | (1.53e-4) | (4.45e-4) | **(0.16)** | (1.45e-4) | (1.65e-4) | **(0.09)** |
| CPMCN | **1.43e-3** | **2.01e-3** | **93.18** | **1.25e-3** | **1.51e-3** | **92.98** | **1.61e-3** | **1.68e-3** | **90.85** |
| | **(2.32e-4)** | **(8.17e-4)** | **(0.16)** | **(1.53e-4)** | **(4.45e-4)** | **(0.16)** | **(1.45e-4)** | **(1.65e-4)** | **(0.09)** |

Table 5: Performance on CIFAR10 under the Dirichlet shift. For each method, the first line is the median of the corresponding metrics among 100 repeating experiments and the second line is the standard derivation of these 100 metrics.The best results are marked as **bold**.

| $\rho$ | 0.01 | | | 0.02 | | | 0.05 | | |
|---|---|---|---|---|---|---|---|---|---|
| Metrics | MSE_EVEN | MSE_PROP | ACC (%) | MSE_EVEN | MSE_PROP | ACC (%) | MSE_EVEN | MSE_PROP | ACC (%) |
| LTF | 3.83e-2 (4.69e-3) | 2.43e-2 (5.04e-3) | 91.55 (0.25) | 3.05e-2 (5.63e-3) | 2.39e-2 (4.92e-3) | 91.50 (0.33) | 2.40e-2 (8.41e-3) | 2.25e-2 (9.36e-3) | 90.93 (0.72) |
| BBSL | 1.52e-3 (1.02e-4) | 1.50e-3 (9.74e-5) | 92.65 (0.06) | 1.55e-3 (1.12e-4) | 1.52e-3 (9.63e-5) | **92.30** (**0.06**) | 1.47e-3 (1.05e-4) | **1.42e-3** (**9.44e-5**) | **91.63** (**0.06**) |
| RLLS | 1.52e-3 (9.94e-5) | 1.50e-3 (9.57e-5) | 92.65 (0.06) | 1.55e-3 (1.12e-4) | 1.52e-3 (9.63e-5) | **92.30** (**0.06**) | 1.47e-3 (1.05e-4) | **1.42e-3** (**9.44e-5**) | **91.63** (**0.06**) |
| ELSA | 2.96e-3 (6.32e-3) | 3.16e-3 (1.42e-3) | 92.65 (0.06) | 2.69e-3 (4.19e-4) | 2.75e-3 (4.56e-4) | **92.30** (**0.06**) | 2.86e-3 (4.10e-4) | 2.76e-3 (4.29e-4) | 91.55 (0.06) |
| EM | **1.32e-3** (**8.50e-5**) | **1.41e-3** (**8.98e-5**) | **92.70** (**0.06**) | **1.41e-3** (**8.34e-5**) | **1.43e-3** (**8.55e-5**) | **92.30** (**0.06**) | **1.44e-3** (**7.99e-5**) | 1.44e-3 (7.86e-5) | **91.63** (**0.06**) |
| CPMCN | **1.32e-3** (**8.49e-5**) | **1.41e-3** (**8.97e-5**) | **92.70** (**0.06**) | **1.41e-3** (**8.33e-5**) | **1.43e-3** (**8.54e-5**) | **92.30** (**0.06**) | **1.44e-3** (**7.99e-5**) | 1.43e-3 (7.86e-5) | **91.63** (**0.06**) |

| $\rho$ | 0.6 | | | 0.8 | | | 0.9 | | |
|---|---|---|---|---|---|---|---|---|---|
| Metrics | MSE_EVEN | MSE_PROP | ACC (%) | MSE_EVEN | MSE_PROP | ACC (%) | MSE_EVEN | MSE_PROP | ACC (%) |
| LTF | 4.13e-3 (4.00e-4) | 3.41e-3 (1.36e-3) | 91.89 (0.08) | 3.18e-3 (1.37e-3) | 1.23e-2 (9.57e-3) | 94.67 (0.08) | 4.71e-3 (1.45e-3) | 2.21e-2 (5.73e-3) | **96.65** (**0.22**) |
| BBSL | 6.33e-3 (1.15e-3) | 1.92e-2 (5.79e-3) | 91.80 (0.06) | 1.12e-2 (2.06e-3) | 4.30e-2 (1.41e-2) | 94.52 (0.07) | 1.19e-2 (2.41e-3) | 6.12e-2 (1.82e-2) | 96.23 (0.06) |
| RLLS | 6.33e-3 (1.15e-3) | 1.92e-2 (5.79e-3) | 91.80 (0.06) | 1.12e-2 (2.06e-3) | 4.30e-2 (1.40e-2) | 94.52 (0.07) | 1.18e-2 (2.41e-3) | 5.70e-2 (1.82e-2) | 96.20 (0.06) |
| ELSA | 2.24e-3 (3.35e-4) | **4.13e-3** (**1.59e-3**) | 91.90 (0.06) | **1.29e-3** (**1.55e-3**) | 3.31e-3 (4.06e-3) | 94.72 (0.09) | 1.29e-3 (2.44e-3) | 5.29e-3 (1.17e-2) | 96.55 (0.08) |
| EM | **2.12e-3** (**3.57e-4**) | 5.63e-3 (1.66e-3) | **91.90** (**0.06**) | 1.69e-3 (2.97e-4) | 6.42e-3 (1.74e-3) | **94.75** (**0.06**) | 1.25e-3 (2.43e-4) | **4.92e-3** (**1.44e-3**) | **96.65** (**0.05**) |
| CPMCN | **2.12e-3** (**3.57e-4**) | 5.62e-3 (1.66e-3) | **91.90** (**0.06**) | 1.68e-3 (2.97e-4) | 6.36e-3 (1.74e-3) | **94.75** (**0.06**) | **1.24e-3** (**2.44e-4**) | 4.97e-3 (1.44e-3) | **96.65** (**0.06**) |

Table 6: Performance on CIFAR10 under the tweak-one shift. For each method, the first line is the median of the corresponding metrics among 100 repeating experiments and the second line is the standard derivation of these 100 metrics. The best results are marked as **bold**.

