# OpenReview forum: "Class Probability Matching with Calibrated Networks for Label Shift Adaption"
_ICLR.cc/2024/Conference — ICLR 2024 poster_

### Official Review · Reviewer_FsKm · 2023-10-30

**Soundness:** 3 good
**Presentation:** 3 good
**Contribution:** 3 good
**Rating:** 8
**Confidence:** 3

**Summary:**

This paper proposes a novel method for domain adaptation in the context of label shift, where label distributions between the source and target domains differ while the conditional feature-label distribution remains the same. The proposed approach, called "class probability matching" (CPM), aims to match class probabilities and significantly improves computational efficiency compared to existing methods. Within the CPM framework, they introduce "class probability matching with calibrated networks" (CPMCN) for target domain classification, supported by a theoretical generalization bound.

**Strengths:**

1. This paper proposes a novel class probability matching method, which is computationally more efficient compared with existing feature distribution matching methods.

2. Theoretical analysis of the generalization bound is obtained for the proposed approach.

3. The algorithm is simple, and the empirical results demonstrate superior performance of CPMCN over existing approaches.

**Weaknesses:**

The theoretical analysis is not intuitive. Maybe some proof sketches in the paper can help the readers understand the results.  For example, why assumption 5.1 is required and what does it mean.

**Questions:**

1.	The core of class probability matching is to derive that solving problem (11) can obtain an estimator of class probability ratio w*, which is utilized to transform the source domain classifier to the target domain classifier. However, the authors do not provide any analysis of the problem (1). Can its optimal solution be obtained by existing algorithms, e.g., BFGS. If the approximate solution is attained, how the approximation influences the generalization bound.
2.	In figure 1, the curves of other compared matching approaches are expected to be shown.
3.	Please verify that the proposed approach is computational more efficient than compared matching approaches quantitatively in the experimental studies.

---

> ### Author Response · Authors · 2023-11-20
> **Rebuttal (part 1/2)**
>
> We appreciate your valuable review and feedback.
>
> **Q1.** The theoretical analysis is not intuitive. Maybe some proof sketches in the paper can help the readers understand the results. For example, why assumption 5.1 is required and what does it mean.
>
> **A1.** We provide proof sketches and intuitive explanations of Theorem 5.4 and Assumption 5.1 in Appendix B.3 of the updated version of the manuscript.
>
>
> **Q2.** The core of class probability matching is to derive that solving problem (11) can obtain an estimator of class probability ratio $w^*$, which is utilized to transform the source domain classifier to the target domain classifier. However, the authors do not provide any analysis of the problem (11). Can its optimal solution be obtained by existing algorithms, e.g., BFGS. If the approximate solution is attained, how the approximation influences the generalization bound.
>
> **A2.** The solution of  problem (11) obtained by BFGS is very close to the optimal solution $\widehat{w}$ of problem (11). To be specific, from Theorem 5.4 we can see that the optimal solution $\widehat{w}$ of the optimization problem (11) is a consistent estimator of the true class probability ratio $w^*$. On the other hand, the experimental results in Table 1 show that the approximate solution of (11) by BFGS, denoted as $\overline{w}$, has a negligible MSE for estimating the true ratio $w^*$. Therefore, by the inequality $\\|\overline{w} - \widehat{w}\\|^2 \leq 2(\\|\overline{w} - w^*\\|^2 + \\|w^* - \widehat{w}\\|^2)$, we can see that the approximate solution $\overline{w}$ converges to the optimal solution $\widehat{w}$ of (11).
>
> Now we answer how the approximation influences the generalization bound. Compared to the original generalization bound shown in Theorem 5.5, we find that the generalization bound of the classifier induced by the approximate solution of (11) has one additional term, the approximation error $\\|\overline{w} - \widehat{w}\\|^2$, where $\overline{w}$ and $\widehat{w}$ denote the approximate  and optimal solution of problem (11), respectively. To be specific, combining Equ. (33), Propositions B.7 and B.9 in the appendix, we can show that the generalization error of the predictor $\overline{q}(y|x)$ induced by the approximate solution $\overline{w}$ can be decomposed into
> $$
> \begin{align*}
> \mathcal{R}_q(\overline{q}(y|x)) - \mathcal{R}_q^*
> \leq 2(\mathcal{R}_p(\widehat{p}(y|x)) - \mathcal{R}_p^*) + \\|\overline{w} - w^*\\|_2^2
> \leq 2(\mathcal{R}_p(\widehat{p}(y|x)) - \mathcal{R}_p^*) + 2\\|\overline{w} - \widehat{w}\\|_2^2 + 2\\|\widehat{w} - w^* \\|_2^2.
> \end{align*}
> $$
> Note that only the second term on the right-hand side, i.e. the approximation error $\\|\overline{w} - \widehat{w}\\|^2$ is related to the approximate solution $\overline{w}$ of (11). The first and third terms are not influenced by the approximate solution $\overline{w}$.
>
> Furthermore, by applying Proposition B.1 and B.6 to bound the first and third term, we get the generalization bound of the classifier $\overline{q}(y|x)$ given by
> $$
> \begin{align*}
> \mathcal{R}_q(\overline{q}(y|x)) - \mathcal{R}_q^* \lesssim 3\bigg(\inf\_{f\in \mathcal{F}}\mathcal{R}_p(f) -\mathcal{R}_p^*+\sqrt{\frac{\mathrm{VC}(\mathcal{F})\log n_p}{n_p}} + 2\sqrt{\frac{\log n_p}{n_p}} + \frac{2\log n_q}{n_q}\bigg)
> \\
> &\quad + 2\\|\overline{w} - \widehat{w}\\|_2^2.
> \end{align*}
> $$
>
>
> We can see that the first term on the right-hand side is the same as the generalization bound of $\widehat{q}(y|x)$ shown in Theorem 5.5 and this term is independent of and not influenced by the approximate solution $\overline{w}$. In addtion, the generalization bound of the predictor $\overline{q}(y|x)$ has the additional second term, which is proportional to the approximation error $\\|\overline{w} - \widehat{w}\\|_2^2$. Combining these two terms indicates that the generalization bound of $\overline{q}(y|x)$ increases at most linearly with the approximation error $\\|\overline{w} - \widehat{w}\\|_2^2$.

---

> > ### Author Response · Authors · 2023-11-20
> > **Rebuttal (part 2/2)**
> >
> > **Q3.** In figure 1, the curves of other compared matching approaches are expected to be shown.
> >
> > **A3.** In Figure 1, we provide the convergence curve of the distance between two class probabilities $\widehat{p}(y)$ and $\widehat{p}_q^w(y)$ in Equ. (11) and the MSE curve of the corresponding class probability ratio estimation. However, other compared methods do not minimize the distance between these two class probabilities to estimate the class probability ratio, so the curve in Figure 1 is specific for our CPMCN algorithm.
> >
> >
> > **Q4.** Please verify that the proposed approach is computational more efficient than compared matching approaches quantitatively in the experimental studies.
> >
> > **A4.** In order to empirically compare with other methods, we present the runnning time of all methods on CIFAR100 under the Dirchlet with $\alpha=0.1$.
> >
> > | CPMCN | KMM | LTF | BBSL | RLLS | ELSA | EM |
> > | ------ | ----- | ------- |------ | ------ | ------ | ------ |
> > | 5.27(0.11) | >2e+5 | 29107(1014) | 4.49(0.07) | 5.55(0.08) | 7.48(0.13) | 8.47(0.05) |
> >
> >
> > The time units in the table are in seconds and inside the bracket is the standard deviation. Given that all compared methods utilize pre-trained network predictors, the time spent on pre-training is not included. We also include the running time of the KMM method (Zhang et al., 2013) in the table, although it does not produce any results even after running for over 200,000 seconds.
> >
> > This table shows that compared with the feature probability matching methods KMM and LTF, our class probability matching method CPMCN has significant computational advantage, which coincides with our complexity analysis. In addition, CPMCN's running time is comparable to that of  prediction matching methods including BBSL, RLLS and ELSA, and is even faster than RLLS and ELSA. Additionally, CPMCN outperforms all prediction matching methods significantly. Furthermore, our method is faster than the EM-based algorithm.

---

> > ### Comment · Reviewer_FsKm · 2023-11-22
> >
> > Thank the authors for the response! My concerns are addressed. The score has been turned from 6 to 8.

---

> > > ### Author Response · Authors · 2023-11-22
> > >
> > > Thank you for your positive feedback!
> > >
> > > Kindly note that the system may not yet reflect your adjusted score (it still shows rating 6).

---

### Official Review · Reviewer_Y2x9 · 2023-10-31

**Soundness:** 4 excellent
**Presentation:** 4 excellent
**Contribution:** 3 good
**Rating:** 8
**Confidence:** 5

**Summary:**

This paper studies the domain adaptation problem under label shift, assuming that the domains' label priors differ while the class-conditional probabilities are the same. Existing algorithms are based on feature probability matching, where the class-conditional probability can be difficult to estimate e.g. using GAN. To cope with this problem, the authors propose a novel matching algorithm based on class probability matching. They proved that the estimated class probability ratio is consistent with previous algorithms and the estimation process enjoys wonderful theoretical guarantees. Experimentally, they show the proposed method achieves better performance than existing algorithms.

**Strengths:**

1. This paper is well-written and easy to follow.

2. The theoretical analysis is thorough and helpful.

3. The experimental results clearly verified the effectiveness and efficiency.

Overall, I believe this is a solid paper and should be accepted.

**Weaknesses:**

1. While I vote for clear acceptance, I think the studied problem is not very significant since they simply consider the label shift problem. Notably, this problem can be regarded as an imbalanced semi-supervised learning problem where the labeled set and the unlabeled set have different classes prior. There have been many empirically strong works that study the imbalanced semi-supervised learning problem [1-4] and I suggest the authors discuss these papers properly.

2. Can the proposed method be equipped with existing imbalanced semi-supervised learning algorithms for improved performance? It won't affect my decision, but I would appreciate it if the authors could provide further extensions, which I believe will enhance the empirical significance of this paper.

3. In Eq (11), the authors directly optimize the class prob ratio by searching the $R^K$ space and finally perform normalization. Can we directly obtain the class prob ratio by constraining it in the normalized space $\sum \hat{w}_k = 1$?

4. There are some typos, e.g. there is an unexpected bracket on page 13 line 4, 'p(k))'.

[1] Kim J, Hur Y, Park S, et al. Distribution aligning refinery of pseudo-label for imbalanced semi-supervised learning[J]. Advances in neural information processing systems, 2020, 33: 14567-14579.

[2] Oh Y, Kim D J, Kweon I S. Daso: Distribution-aware semantics-oriented pseudo-label for imbalanced semi-supervised learning[C]//Proceedings of the IEEE/CVF Conference on Computer Vision and Pattern Recognition. 2022: 9786-9796.

[3] Wei C, Sohn K, Mellina C, et al. Crest: A class-rebalancing self-training framework for imbalanced semi-supervised learning[C]//Proceedings of the IEEE/CVF conference on computer vision and pattern recognition. 2021: 10857-10866.

[4] Fan Y, Dai D, Kukleva A, et al. Cossl: Co-learning of representation and classifier for imbalanced semi-supervised learning[C]//Proceedings of the IEEE/CVF conference on computer vision and pattern recognition. 2022: 14574-14584.

**Questions:**

Is the network fixed and will not be fine-tuned on the target domain?

BTW, I accidentally clicked the check box of 'First Time Reviewer' while I've been an experienced reviewer in the ICLR community.

---

> ### Author Response · Authors · 2023-11-20
> **Rebuttal**
>
> We appreciate your valuable review and feedback.
>
> **Q1.** While I vote for clear acceptance, I think the studied problem is not very significant since they simply consider the label shift problem. Notably, this problem can be regarded as an imbalanced semi-supervised learning problem where the labeled set and the unlabeled set have different classes prior. There have been many empirically strong works that study the imbalanced semi-supervised learning problem [1-4] and I suggest the authors discuss these papers properly.
>
> **A1.** Label shift adaptation and imbalanced semi-supervised learning both consider the class probability shift, but have different learning goals. To be specific, both, label shift adaptation and imbalanced semi-supervised learning, acknowledge the challenges posed by changes in the class distribution. However, the difference is that label shift adaptation focuses on the differences in the class distributions between source and target domain. It aims to achieve high classification accuracy on the unlabeled target domain samples. In contrast, imbalanced semi-supervised learning emphasizes leveraging both, imbalanced labeled and unlabeled data, to achieve high accuracy on balanced test data, thereby focusing on the accuracy for minority classes.
>
> In the aspect of algorithm, our method CPMCN for class probability ratio estimation can be applied to an imbalanced semi-supervised learning problem for estimating the class probability of unlabeled data, which is commonly used to improve the quality of the pseudo-labels [1-3] or features [4] of the unlabeled data. On the other hand, the idea of using the estimated class probability to refine the pseudo-labels of unlabeled data can be used for label shift adaption problems to further improve the classification accuracy on the target domain data.
>
> We also discuss these papers in our paper.
>
>
> **Q2.** Can the proposed method be equipped with existing imbalanced semi-supervised learning algorithms for improved performance? It won't affect my decision, but I would appreciate it if the authors could provide further extensions, which I believe will enhance the empirical significance of this paper.
>
> **A2.** Yes, to our mind a combination of CPMCN and an imbalanced semi-supervised learning algorithms is possible. For example, we can potentially combine our algorithm CPMCN and the technique called "distribution aligning refinery of pseudo-label"(DARP) in [1] for imbalanced semi-supervised learning. To be specific, after getting the estimated class probability $\widehat{q}(y) := \widehat{w}_y \widehat{p}(y)$ and the predictor $\widehat{q}(y|x)$ by our CPMCN, we will use DARP to refine the label prediction $\widehat{q}(y|X_i)$ for the target domain data $X_i \in D_q^u$ so that the class probability of the refined label prediction aligns with the estimated class probability $\widehat{q}(y)$. Due to time constraints, we are still conducting experimental exploration on this new combination algorithm. If we find that it has a significant empirical advantage over the original CPMCN in most label shift cases, we will also report the results of this method named  "CPMCN + DARP" in the updated version of our article.
>
>
> **Q3.** In Eq (11), the authors directly optimize the class prob ratio by searching the space and finally perform normalization. Can we directly obtain the class prob ratio by constraining it in the normalized space?
>
> **A3.** Yes, it is potentially feasible to directly solve the class prob ratio when applying sequential quadratic programming or the augmented lagrangian method. In our current implementation, we use BFGS equipped with normalization since the solution of BFGS almost satisfies the constraint condition so the normalization is a simple and effective way for correction.
>
>
> **Q4.** There are some typos, e.g. there is an unexpected bracket on page 13 line 4, 'p(k))'.
>
> **A5.** Thanks for noting. We corrected the indicated typo and we will carefully check the manuscript again before publication.
>
> **Q5.** Is the network fixed and will not be fine-tuned on the target domain?
>
> **A5.** Yes, the calibrated network is trained only with the source domain data and we did not fine-tune it on the target domain.

---

### Official Review · Reviewer_t6VC · 2023-10-31

**Soundness:** 3 good
**Presentation:** 3 good
**Contribution:** 3 good
**Rating:** 8
**Confidence:** 3

**Summary:**

The paper presents a method which addresses the label shift problem. For dealing with label shift, it rewrites the target class probability as a tractable function with unknown parameters, averaged over the target feature distribution. The parameters are optimized by the BFGS algorithm and serve as weights to construct prediction scores on a target domain. The important condition is that the underlying classifier should be calibrated.

**Strengths:**

While the idea somewhat resembles the probability matching framework, it considers the problem from a different angle of matching class probabilities. The paper is well written and easy to follow. It contains necessary theoretical grounding and experiments which show method’s superiority over the baselines. The method has an advantage in terms of the runtime complexity, and demonstrates better accuracy results.

**Weaknesses:**

1. While the computational complexity is considered as one of the main advantages of the method, the paper is missing some experimental studies on the runtime. At what point (target domain size) the method becomes computationally prohibitive? How does that compare empirically with the other methods?
2. A minor comment on writing: introducing “Calibrated networks” as a part of the method might suggest that there is some novelty in designing a proper mechanism of calibration for the label shift scenario. In fact, calibrated networks are used off the shelf as a necessary ingredient of the method to improve the estimation of p(y|x).

**Questions:**

The method currently doesn’t allow for a minibatch training; I’m wondering whether the authors thought about adapting the algorithm to handle minibatching?

---

> ### Author Response · Authors · 2023-11-20
> **Rebuttal**
>
> We appreciate your valuable review and feedback.
>
> **Q1.** While the computational complexity is considered as one of the main advantages of the method, the paper is missing some experimental studies on the runtime. At what point (target domain size) the method becomes computationally prohibitive? How does that compare empirically with the other methods?
>
> **A1.** From the perspective of the computational complexity, the CPMCN's running time scales linearly with the sample size. To verify this empirically, we provide the running time of our CPMCN for the different target domain sizes $n_q = 2000, 4000, 8000$.
>
> | $n_q$ | 2000 | 4000 | 8000 |
> | ------ | ------- |------ | ------ |
> | CPMCN | 5.27(0.11) | 10.95 (0.29) | 23.40 (0.76) |
>
> The time units in the table are in seconds and inside the bracket is the standard deviation. This verifies our complexity analysis and thus our algorithm can deal with large-scale target domain datasets in acceptable running time.
>
> In order to empirically compare with other methods, we present the running time of all methods on CIFAR100 under the Dirichlet with $\alpha=0.1$.
>
> | CPMCN | KMM | LTF | BBSL | RLLS | ELSA | EM |
> | ------ | ----- | ------- |------ | ------ | ------ | ------ |
> | 5.27(0.11) | >2e+5 | 29107(1014) | 4.49(0.07) | 5.55(0.08) | 7.48(0.13) | 8.47(0.05) |
>
> Given that all compared methods utilize pre-trained network predictors, the time spent on pre-training is not included. We also include the running time of the KMM method (Zhang et al., 2013) in the table, although it does not produce any results even after running for over 200,000 seconds.
>
> This table shows that compared with the feature probability matching methods KMM and LTF, our class probability matching method CPMCN has significant computational advantage, which coincides with our complexity analysis. In addition, compared with prediction matching methods including BBSL, RLLS and ELSA, CPMCN's running time are comparable and even faster than RLLS and ELSA. Additionally, CPMCN outperforms all prediction matching methods significantly. Furthermore, our method is faster than the EM-based algorithm.
>
> We also included this running time comparison table in our updated paper.
>
> **Q2.** A minor comment on writing: introducing “Calibrated networks” as a part of the method might suggest that there is some novelty in designing a proper mechanism of calibration for the label shift scenario. In fact, calibrated networks are used off the shelf as a necessary ingredient of the method to improve the estimation of p(y|x).
>
> **A2.** As the reviewer pointed out, the statement "introducing calibrated networks as a part of the method" could be misleading. In reality, our intention is to convey that our newly proposed Class Probability Matching (CPM) framework allows calibrated networks to be employed within our algorithm. Put differently, our CPM seamlessly integrates calibrated networks, resulting in promising empirical results. To prevent any potential misinterpretation, we have revised the title and introduction of Section 4 in the paper to explicitly convey the incorporation of the CPM framework and calibrated networks.
>
>
> **Q3.** The method currently doesn’t allow for a minibatch training; I’m wondering whether the authors thought about adapting the algorithm to handle minibatching?
>
> **A3.** While the current method does not explicitly support minibatching, considering the incorporation of minibatch training could be beneficial, especially in scenarios where the target domain data size is large. Minibatch training allows for the iterative update of the estimation of class probability ratios, which can enhance computational efficiency and improve the scalability.

---

### Official Review · Reviewer_PKvz · 2023-11-01

**Soundness:** 3 good
**Presentation:** 3 good
**Contribution:** 2 fair
**Rating:** 5
**Confidence:** 4

**Summary:**

This paper introduces a solution for the label shift adaptation problem: a matching framework called "Class Probability Matching" (CPM). CPM offers the same theoretical guarantees as the previous framework (feature probability matching framework) and addresses the issue of its low computational efficiency. Inspired by the CPM framework, the authors further develop the CPMCN algorithm, utilizing calibrated neural networks for classification in the target domain, and demonstrate the benefits of calibrated neural networks from the theoretical perspective. In experiments, CPMCN outperforms other matching methods as well as EM-based algorithms.

**Strengths:**

1. Paper is well written and easy to follow;
2. The authors compare, analyze, and make improvements to existing frameworks, and these improvements are well-founded;
3. The proposed method is simple but effective.

**Weaknesses:**

1. The authors emphasize that the proposed algorithm has made significant improvements in terms of computational efficiency, reducing the computational complexity from O(n_p^3) to O(n_qK^2)  . However, no related experimental evidence has been provided to support this
claim;
2. While this paper is the first to estimate marginal probability ratios from a class probabilistic perspective, such methods have already been explored extensively in the long-tailed domain, especially in approaches like logit adjustment, as seen in prior works such as the one in [1]. I
believe it is essential for the authors to conduct some research and comparisons in this regard;
3. Previous studies [2] have already demonstrated that calibration can indeed enhance performance, so the calibrated network mentioned in the paper is not very novel.

[1] Generalized Logit Adjustment: Calibrating Fine-tuned Models by Removing Label Bias in Foundation Models. // Main Track, Conference on Neural Information Processing Systems. NeurIPS 2023.
[2] Alexandari A, Kundaje A, Shrikumar A. Maximum likelihood with bias-corrected calibration is hard-to-beat at label shift adaptation[C]//International Conference on Machine Learning. PMLR,
2020: 222-232.

**Questions:**

1. Could the authors provide a brief explanation of the calibrated network mentioned in the paper?
2. Why do the experiments in the 'Performance on MNIST under the tweak-one shift' have so many identical results? Even the standard deviation is the same for the accuracy metric at  \ro=0.02.

---

> ### Author Response · Authors · 2023-11-20
> **Rebuttal (part 1/2)**
>
> We appreciate your valuable review and feedback.
>
> **Q1.** The authors emphasize that the proposed algorithm has made significant improvements in terms of computational efficiency, reducing the computational complexity from $O(n_p^3)$ to $O(n_qK^2)$. However, no related experimental evidence has been provided to support this claim.
>
> **A1.** To provide experimental evidence to show our computational efficiency over the Feature Probability Matching (FPM) method, we compare the running time of our method, CPMCN, with two compared FPM-based methods named kernel mean matching (KMM) (Zhang et al., 2013) and LTF (Guo et al., 2020). The running time comparison on CIFAR100 under the Dirichlet shift with $\alpha=0.1$ is presented in the following table.
>
> | CPMCM | KMM | LTF |
> | ------ | ------- | ----- |
> | 5.27(0.11) | >2e+5 | 29107(1014) |
>
>
> The time units in the table are in seconds and inside the bracket is the standard deviation. Given that all compared methods utilize pre-trained network predictors, the time spent on pre-training is not included.
>
> In the table above, KMM, as a feature probability matching (FPM) approach, fails to produce results even after running for over 200,000 seconds, confirming its substantial computational complexity of $O(n_p^3)$. Moreover, another FPM-based method, LTF, is also significantly slower than our CPMCN approach based on class probability matching (CPM). This empirically support our discussion on the computational advantage of our CPM over the compared FPM.
>
>
> **Q2.** While this paper is the first to estimate marginal probability ratios from a class probabilistic perspective, such methods have already been explored extensively in the long-tailed domain, especially in approaches like logit adjustment, as seen in prior works such as the one in [1]. I believe it is essential for the authors to conduct some research and comparisons in this regard;
>
> **A2.** Both, our label shift paper and the paper [1] on the long-tailed problem, involve the common task of estimating class probabilities. However, it is important to note that these two problems are considered against distinct settings, meaning that the available or observable information differs. Therefore, the method in [1] cannot be directly used to estimate class probabilities in the label shift problem. In the following, we explain this in detail.
>
> In the context of the label shift problem, our goal is to estimate the class probability ratio $q(y)/p(y)$, or the class probability $q(y)$ of the target domain. The challenge we face stems from the lack of information about $q(y)$. Specifically, in the target domain, we only have unlabeled data from the feature distribution $q(x)$.
>
> In [1], a similar need arises to estimate the class probability $q(y)$ of the pre-training distribution $q(x,y)$. However, a crucial distinction from the label shift setting is that we have a zero-shot predictor $q(y|x)$ trained on the pre-training data but we cannot observe neither labeled nor unlabeled pre-training data. In [1], the availability of $q(y|x)$ allows for the utilization of labeled balanced data, in conjunction with the logit adjustment algorithm, to estimate the class probability $q(y)$ of the pre-training distribution.
>
> In summary, both of the goals are to estimate the class probability $q(y)$ under the label shift assumption. The difference is for label shift adaptation $q(y|x)$ is unaccessible, but $q(x)$ can be directly estimated from the observations, while, in [1], we have the pre-trained model $q(y|x)$, but $q(x)$ is not accessible. Therefore, these two settings are different and complementary. Thus, the method of class probability estimation in [1] cannot be directly applied to the estimation of the class probability $q(y)$ in the target domain. This also demonstrates the value and originality of our class probability matching method within the label shift adaptation framework.
>
> We also dicuss [1] for long-tailed problem in our paper.
>
>
> **Q3.** Previous studies [2] have already demonstrated that calibration can indeed enhance performance, so the calibrated network mentioned in the paper is not very novel.
>
> **A3.** Indeed, previous studies [2] have already shown that calibration can improve performance. Nonetheless, one of our novel contributions is to ENABLE existing calibration networks to be exploited in the newly proposed class probability matching framework, instead of the calibration network itself. By contrast, to the best of our knowledge, previous matching methods for label shift adaption do not use calibrated networks in their papers. Empirical results also show that our new combination of class probability matching and calibration networks are crucial to obtain satisfactory empirical results. Moreover, from the theoretical perspective, we demonstrate the advantages of using calibrated networks in our CPMCN via bias-variance error analysis, which is a novel theoretical contribution.

---

> ### Author Response · Authors · 2023-11-20
> **Rebuttal (part 2/2)**
>
> **Q4.** Could the authors provide a brief explanation of the calibrated network mentioned in the paper?
>
> **A4.** The calibration network used in our CPMCN is constructed by bias-corrected temperature scaling(BCTS) [2]. To be specific, let $z(x)$ be a vector of the original softmax logits computed on the input $x$. BCTS introduces a temperature scale $T$ and bias vector $b := (b_k)\_{k\in [K]}$ to get the calibrated predictor $\widehat{p}(y|x) := \exp(z(x)/T+b_y)/(\sum_{k\in [K]} \exp(z(x)/T+b_k))$ for $y\in [K]$, where $T$ and $b$ are both optimized with respect to the negative log-likelihood on the validation data.
>
> The above content will be added to Section 4 of our paper for illustrating the calibrated networks that we use in CPMCN.
>
> **Q5.** Why do the experiments in the 'Performance on MNIST under the tweak-one shift' have so many identical results? Even the standard deviation is the same for the accuracy metric at $\rho=0.02$.
>
> **A5.** Due to the relatively simple nature of MNIST classification, several compared methods can attain the near-optimal performance level, even under the severe label shift conditions. In essence, these methods consistently yield high accuracy results across almost all repeated experiments, resulting in identical median and standard deviation values for accuracy on MNIST. In contrast, when dealing with the CIFAR100 dataset, which features a greater number of classes and a more complex structure, the performance advantage of our method compared to other methods becomes significant.
>
> [1] Generalized Logit Adjustment: Calibrating Fine-tuned Models by Removing Label Bias in Foundation Models.// Main Track, Conference on Neural Information Processing Systems. NeurIPS 2023.
>
> [2] Alexandari A, Kundaje A, Shrikumar A. Maximum likelihood with bias-corrected calibration is hard-to-beat at label shift adaptation[C]//International Conference on Machine Learning. PMLR, 2020: 222-232.

---

### Public Comment · ~Yunrui_Zhang1 · 2024-08-22
**Code for CPMCN**

Dear Authors,

I would like to kindly inquire if you could make the code for this paper publicly available. It would greatly benefit reproducibility and facilitate future comparisons.

Thank you very much in advance.

---

> ### Public Comment · ~Hongwei_Wen1 · 2024-09-06
> **Code link**
>
> Please find codes via the link https://github.com/hongwei-wen/CPMCN-for-label-shift

---

### Meta-Review · Area_Chair_ia5Z · 2023-12-07

**Metareview:**

This paper introduces a novel solution for the label shift adaptation problem using Class Probability Matching. The idea is straightforward. CPM originally was designed for feature probability matching. This paper extends that to the label space and addresses the issue of its low computational efficiency.  Authors further develop the CPMCN using calibrated neural networks for classification in the target domain, and provide the theoretical benefit of the calibration networks. All reviewers agree that the problem is important, the method proposed is solid, and the presentation is very clear.

**Justification For Why Not Higher Score:**

Several reviewers mentioned improvements the paper can further benefit from.  The idea is also relatively simple and built on a well-known previous work.

**Justification For Why Not Lower Score:**

Most reviewers recognize the contribution of this paper. It is definitely above the acceptance line.

---

### Decision · Program_Chairs · 2024-01-16

Accept (poster)